# Genome and epigenome wide studies of neurological protein biomarkers in the Lothian Birth Cohort 1936

Robert F. Hillary [1], Daniel L. McCartney [1], Sarah E. Harris[2,3], Anna J. Stevenson[1], Anne Seeboth[1], Qian Zhang[4], David C. Liewald [2], Kathryn L. Evans[1,2], Craig W. Ritchie[5], Elliot M. Tucker-Drob[6,7], Naomi R. Wray [4], Allan F. McRae[4], Peter M. Visscher [4], Ian J. Deary[2,3] & Riccardo E. Marioni[1,2]

Although plasma proteins may serve as markers of neurological disease risk, the molecular mechanisms responsible for inter-individual variation in plasma protein levels are poorly understood. Therefore, we conduct genome- and epigenome-wide association studies on the levels of 92 neurological proteins to identify genetic and epigenetic loci associated with their plasma concentrations (n = 750 healthy older adults). We identify 41 independent genome-wide significant ($P < 5.4 \times 10^{-10}$) loci for 33 proteins and 26 epigenome-wide significant ($P < 3.9 \times 10^{-10}$) sites associated with the levels of 9 proteins. Using this information, we identify biological pathways in which putative neurological biomarkers are implicated (neurological, immunological and extracellular matrix metabolic pathways). We also observe causal relationships (by Mendelian randomisation analysis) between changes in gene expression (DRAXIN, MDGA1 and KYNU), or DNA methylation profiles (MATN3, MDGA1 and NEP), and altered plasma protein levels. Together, this may help inform causal relationships between biomarkers and neurological diseases.

[1] Centre for Genomic and Experimental Medicine, Institute of Genetics and Molecular Medicine, University of Edinburgh, Edinburgh EH4 2XU, UK. [2] Centre for Cognitive Ageing and Cognitive Epidemiology, University of Edinburgh, Edinburgh EH8 9JZ, UK. [3] Department of Psychology, University of Edinburgh, Edinburgh EH8 9JZ, UK. [4] Institute for Molecular Bioscience, University of Queensland, Brisbane, QLD 4072, Australia. [5] Edinburgh Dementia Prevention, Centre for Clinical Brain Sciences, University of Edinburgh, Edinburgh EH16 4UX, UK. [6] Department of Psychology, The University of Texas at Austin, Austin, TX 78712, USA. [7] Population Research Center, The University of Texas at Austin, Austin, TX 78712, USA. Correspondence and requests for materials should be addressed to R.E.M. (email: riccardo.marioni@ed.ac.uk)

Plasma proteins execute diverse biological processes and aberrant levels of these proteins are implicated in various disease states. Consequently, plasma proteins may serve as biomarkers, contributing to individual disease risk prediction and personalised clinical management strategies[1]. Identifying circulating biomarkers is of particular importance in neurological disease states in which access to diseased neural tissue in vivo is almost impossible. Furthermore, in neurodegenerative disorders, symptomatology may appear in only advanced clinical states, necessitating early detection and intervention[2]. Elucidating the factors which underpin inter-individual variation in plasma protein levels can inform disease biology and also identify proteins with likely causal roles in a given disease, augmenting their value as predictive biomarkers. Indeed, studies have characterised genetic variants (protein quantitative trait loci; pQTLs) associated with circulating protein levels and utilised such genetic information to identify proteins with causal roles in conditions such as cardiovascular diseases[3–5]. However, studies which have aimed to examine the genetic determinants of neurology-related protein levels in human plasma are limited[6–8]. Furthermore, few studies have combined genetic with epigenetic data to provide an additional layer of information regarding the molecular mechanisms responsible for regulating blood protein levels[9]. Therefore, the goal of the present study was to characterise genetic and epigenetic (using DNA methylation) factors associated with putative neurology-related protein biomarkers in order to identify potential molecular determinants which regulate their plasma levels.

Here, genome-wide and epigenome-wide association studies (GWAS/EWAS) are carried out on the plasma levels of 92 neurological proteins in 750 relatively healthy older adults from the Lothian Birth Cohort 1936 study (mean age: 73; levels adjusted for age, sex, population structure and array plate; hereafter simply referred to as protein levels). These proteins represent the Olink® neurology panel and encompass a mixture of proteins with established links to neurobiological processes (such as axon guidance and synaptic function) and neurological diseases (such as Alzheimer's disease (AD)), as well as exploratory proteins with roles in processes including cellular regulation, immunology, and development. Following the identification of genotype-protein associations (pQTLs), functional enrichment analyses are performed on independent pQTL variants. Upon identification of epigenetic factors associated with protein levels, tissue specificity and pathway enrichment analyses are conducted to reveal possible biological pathways in which neurological proteins are implicated. Protein QTL data are integrated with publicly available expression QTL data to probe the molecular mechanisms which may modulate circulating protein levels. Finally, GWAS summary data for proteins and disease states are integrated using two-sample Mendelian Randomisation (MR) to determine whether selected proteins are causally associated with neurological disease states.

## Results

**Genome wide study of neurological protein biomarkers.** For the GWAS, a Bonferroni $P$ value threshold of $5.4 \times 10^{-10}$ (genome-wide significance level: $5.0 \times 10^{-8}/92$ proteins) was set. The GWAS analysis in 750 older adults identified 2734 significant SNPs associated with 37 proteins (Supplementary Data 1). Conditional and joint analysis (GCTA-COJO) resulted in the identification of 41 conditionally significant pQTLs associated with the levels of 33 proteins ($P < 5.4 \times 10^{-10}$; Fig. 1a; Supplementary Data 2). Notably, while genome-wide significant associations were present for an additional four proteins (Alpha-2-MRAP, CD38, MRS1 and SMPD1), the conditional $P$ value for these

signals following COJO ($n = 1$ independent signal per protein) did not surpass the Bonferroni-corrected threshold of $P < 5.4 \times 10^{-10}$. Of these 41 variants, 36 (87.8%) were *cis* pQTLs (SNP within 10 Mb of the transcription start site (TSS) of the gene) and 5 (12.2%) were *trans* variants. Three of the five *trans* variants were located on chromosomes distinct from their respective Olink® gene. Furthermore, *cis* only associations were present for 28/33 proteins (84.8%), compared to *trans* only associations for 3/33 proteins (9.1%). Two proteins (6.1%) were associated with both *cis* and *trans* pQTLs (CD200R and Siglec-9). For all conditionally significant *cis* pQTLs associated with a given protein, the pQTL with the lowest $P$ value was denoted as the sentinel variant ($n = 30$). The significance of *cis* associations decreased as the distance of the sentinel variant from the TSS increased (Fig. 1b).

The minor allele frequency of independent pQTL variants was inversely associated with effect size (Fig. 1c). Notably, this association may be, in part, due to ascertainment bias as rarer variants (with lower minor allele frequencies) must have large effect sizes to attain the same level of power as more common variants. Independent pQTLs explained between 5.1% (rs12139487; DRAXIN; $P = 4.38 \times 10^{-10}$) and 52.5% (rs6938061; MDGA1; $P = 1.39 \times 10^{-87}$) of the phenotypic variance in plasma protein levels (Supplementary Data 2; Fig. 1d). The majority of pQTL variants were located in intergenic and intronic regions (Supplementary Data 2; Fig. 1e). The number of independent loci associated per protein is shown in Fig. 1f. One *trans* conditionally significant variant (rs4857414) was shared between Siglec-9 and CD200R. This variant was annotated to the *ST3GAL6-AS1* gene. Figure 2 demonstrates the effect of genetic variation at the most significant *cis* pQTL (rs6938061; MDGA1) and *trans* pQTL (rs4857414; Siglec-9) on protein levels.

We also used an alternative method, FUMA (FUnctional Mapping and Annotation) to find independent pQTLs. This approach identified 62 significant pQTLs associated with the levels of 37 proteins (90.3% *cis* and 9.7% *trans* effects; Bonferroni-corrected level of significance: $P < 5.4 \times 10^{-10}$) (Supplementary Data 3). In contrast to GCTA-COJO, FUMA retains the most significant pQTL to identify independent signals through linkage disequilibrium (LD)-based pruning; therefore, variants were identified for all 37 proteins. Seven independent pQTLs associated with the levels of 6 proteins were found using both approaches, whereas the remaining SNPs identified by COJO for a given protein were located within the same locus as corresponding SNPs identified by FUMA (overlapping SNPs highlighted in Supplementary Data 2). In addition, we calculated a measure of LD ($r^2$) between SNPs which were discordant between COJO and FUMA. As 7 independent pQTLs were identified by both methods, this left 34 (41–7) discordant SNPs from COJO and 55 (62–7) discordant SNPs from FUMA. Furthermore, as some proteins contained multiple QTLs, this resulted in 74 SNP-SNP comparisons between COJO and FUMA. SNPs which exhibited an $r^2$ coefficient > 0.75 were considered to show evidence of replication (through LD) between both methods. In total, 27 COJO SNP-FUMA SNP comparisons exhibited an $r^2$ > 0.75. This consisted of 26 unique SNPs identified by COJO and encompassed 24 proteins (Supplementary Table 1).

**Colocalisation of *cis* pQTLs with *cis* eQTLs.** Of the 30 sentinel *cis* pQTL variants, 12 (40.0%) were *cis* eQTLs for the same gene in blood tissue. For 3/12 proteins (DRAXIN, KYNU and MDGA1), there was strong evidence (posterior probability (PP) > 0.95) for colocalisation of *cis* pQTLs and *cis* eQTLs and for 2 proteins, LAIR-2 and SIGLEC9, there was weaker evidence (PP > 0.75) for colocalisation. For 5/13 proteins, there was strong evidence

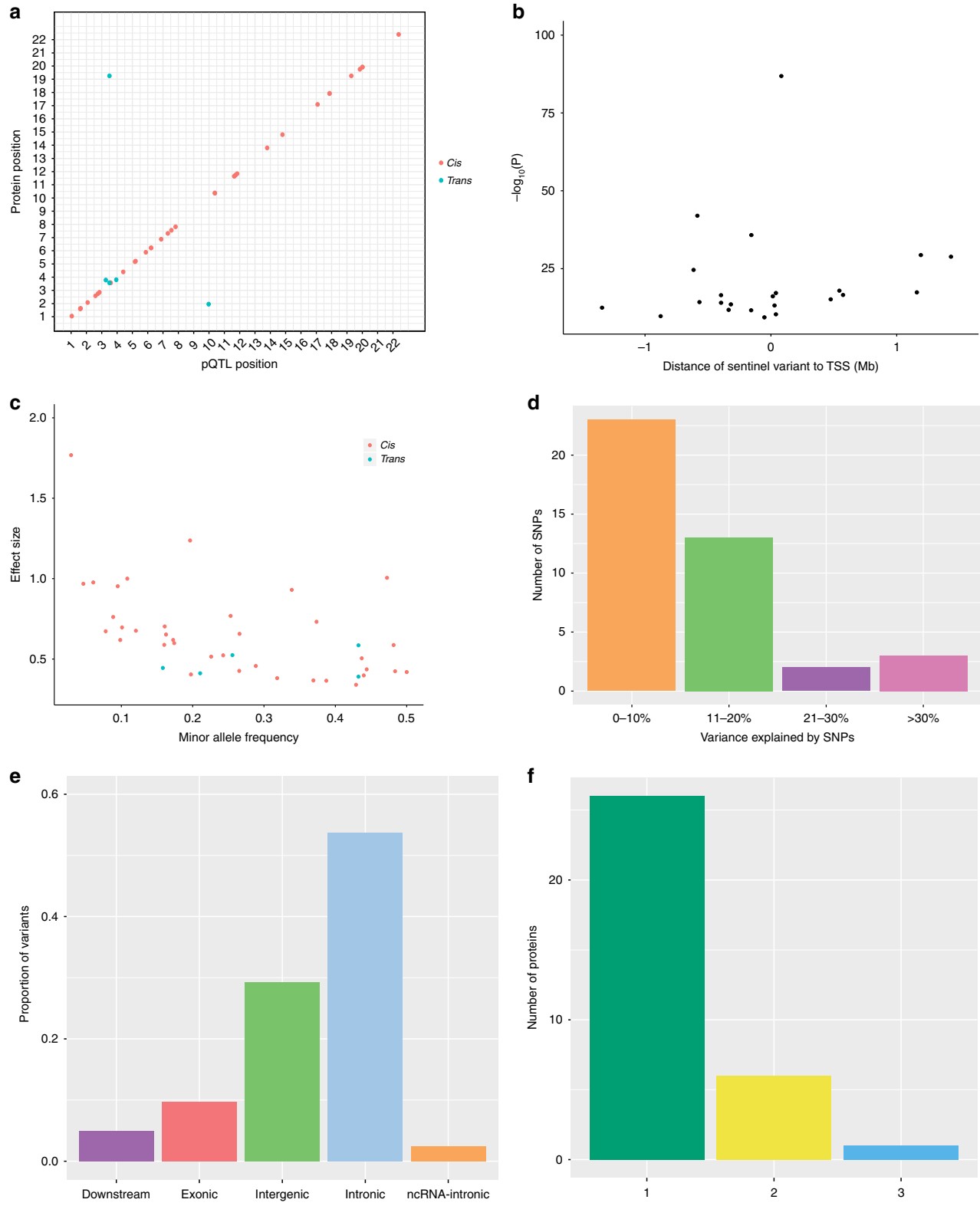

**Fig. 1** Genome-wide association study of neurological protein biomarkers. **a** Chromosomal locations of pQTLs. The *x*-axis represents the chromosomal location of conditionally significant *cis* and *trans* SNPs associated with the levels of Olink® neurology proteins. The *y*-axis represents the position of the gene encoding the associated protein. *Cis* (red circles); *trans* (blue circles). **b** Significance of sentinel *cis* variants versus distance of variants from the gene transcription start site. **c** Absolute effect size (per standard deviation of difference in protein level per effect allele) of conditionally significant pQTLs versus minor allele frequency. *Cis* (red circles); *trans* (blue circles). **d** Variance in protein levels explained by conditionally significant pQTLs. It is important to note that these estimates may be inflated owing to winner's curse or over-fitting in the discovery GWAS. **e** Classification of pQTL variants by function as defined by functional enrichment analysis in FUMA. **f** Number of conditionally significant pQTL variants per Olink® neurology protein

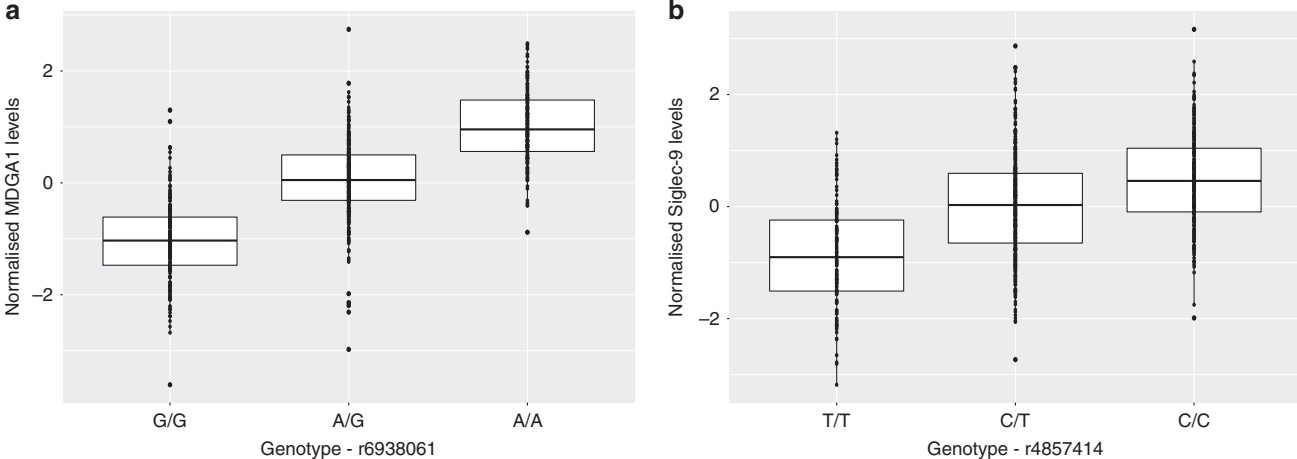

**Fig. 2** Effect of genetic variation on neurological protein levels. **a** Box plot of MDGA1 levels as a function of genotype (rs6938061, effect allele: A, other allele: G, beta = 1.00, se = 0.05). **b** Box plot of Siglec-9 levels as a function of genotype (rs4857414, effect allele: T, other allele: C, beta = −0.58, se = 0.05). Centre line of boxplot: median, bounds of box: first and third quartiles and tips of whiskers: minimum and maximum

(PP > 0.95) for two distinct causal variants affecting transcript and protein levels in the locus. For CTSC, there was weaker evidence (PP > 0.75) for two separate causal signals affecting gene expression and plasma protein levels within the locus. Finally, for CLM-6, there was weak evidence (PP > 0.75) for a causal variant affecting gene expression, but not protein levels, within the locus (Supplementary Table 2).

For the 3 proteins with strong evidence in favour of a shared causal variant for gene expression and plasma protein levels, two-sample MR was performed to test for a causal association between perturbations in gene expression (using data from eQTLGen Consortium) and plasma protein levels (using our GWAS data). Pruned *cis* protein and expression QTL variants (LD $r^2 < 0.1$) were used as instrumental variables for the bidirectional MR analyses. For each trait, the intercept from MR Egger regression was non-significant, which does not suggest strong evidence for directional pleiotropy (DRAXIN: $P = 0.82$; MDGA1: $P = 0.38$; KYNU: $P = 0.36$). For 2 proteins, variation in gene expression was causally associated with plasma protein levels (Inverse variance-weighted method; MDGA1: beta = 0.99, se = 0.49, $P = 0.02$; KYNU: beta = 1.05, se = 0.22, $P = 2.2 \times 10^{-6}$). We did not observe a causal relationship between gene expression of DRAXIN and altered plasma protein levels (Inverse variance-weighted method; beta = −0.98, se = 0.62, $P = 0.10$); however, we did observe a causal relationship between DRAXIN plasma protein levels and changes in gene expression (beta = −0.72, se = 0.07, $P = 1.2 \times 10^{-23}$).

**Epigenome wide study of neurological protein biomarkers**. For the EWAS, a Bonferroni $P$ value threshold of $3.9 \times 10^{-10}$ (genome-wide significance level: $3.6 \times 10^{-8}/92$ proteins) was set[10] and analyses were performed using limma, a linear-model-based method. We identified 26 genome-wide significant CpG sites associated with the levels of 9 neurological proteins ($P < 3.9 \times 10^{-10}$). Of these associations, 17 were *cis* effects (65.4%) and 9 associations were *trans* effects (35.6%; with 6 *trans* variants located on chromosomes distinct from their respective Olink® gene) (Fig. 3; Supplementary Table 3). As an additional analysis, we performed a mixed-linear-model approach termed OSCA (OmicS-data-based Complex trait Analysis)-MOMENT. OSCA has been recently shown to identify fewer spurious signals than other methods (including linear regression) (Zhang et al.[41]). Of the 9 proteins with genome-wide significant CpG sites identified using limma ($n = 26$ CpG sites), 8 proteins were also shown to

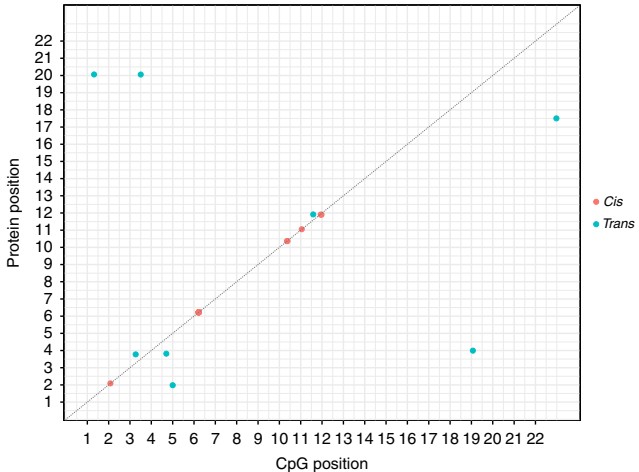

**Fig. 3** Genomic locations of CpG sites associated with differential neurological protein levels. The *x*-axis represents the chromosomal location of CpG sites associated with the levels of Olink® neurology biomarkers. The *y*-axis represents the position of the gene encoding the associated protein. Notably, *cis* CpG sites ($n = 17$) identified by our EWAS on protein levels lay within the same cluster for a given protein. Some of these CpG sites lay too close to discriminate, resulting in the appearance of 5 *cis* CpG clusters in this figure

have genome-wide significant associations using OSCA ($n = 23$ CpG sites; 14 *cis* (60.9%) and 9 *trans* (39.1%) associations). Indeed, only CRTAM failed to show a Bonferroni-corrected significant association using OSCA when compared to limma. Furthermore, of the 23 CpG sites identified using OSCA, 19/23 CpGs (82.6%) were also identified by EWAS performed using limma showing a strong overlap between both methods (Supplementary Table 4).

Three proteins exhibited both genome-wide significant SNP and CpG site associations: MATN3, MDGA1, and NEP (Fig. 4). For MATN3, the *cis* pQTL identified in this study (rs3731663) has previously been identified as a methylation QTL (mQTL) for the single *cis* CpG site associated with MATN3 levels identified by our EWAS (cg24416238)[11]. Similarly, the 2 *cis* pQTLs for differential blood MDGA1 concentrations in our study have been significantly associated with methylation levels of *cis* CpG

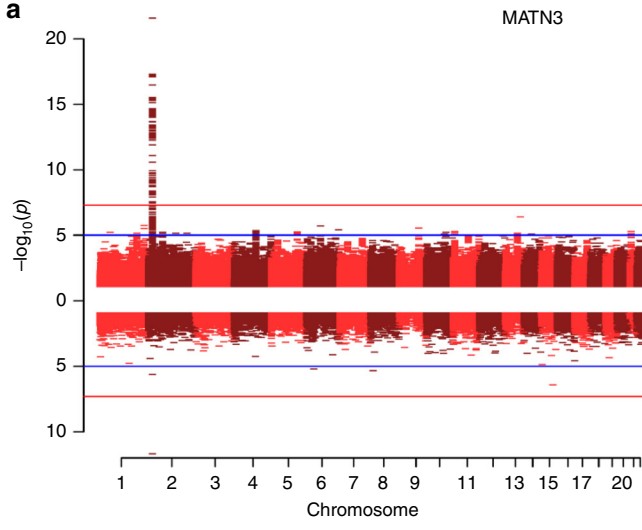

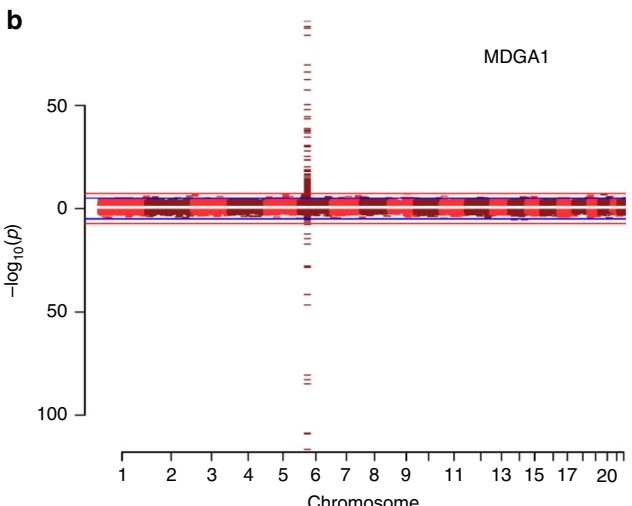

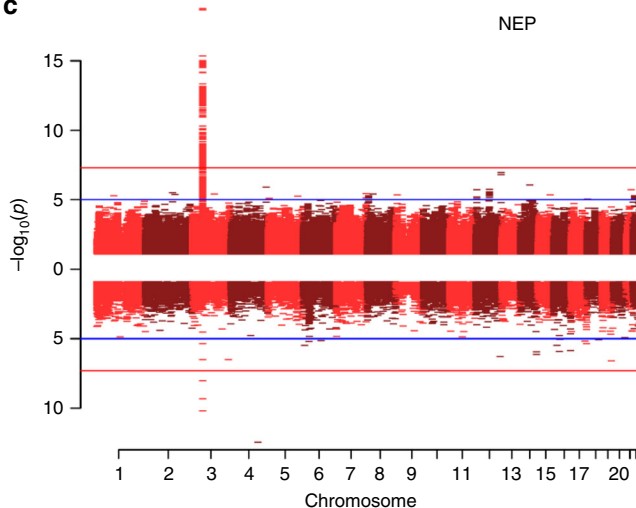

**Fig. 4** Miami plots of three neurological proteins with both genome-wide significant SNP and genome-wide significant CpG associations. The top half of the plot (skyline) shows the results from the GWAS on protein levels, whereas the bottom half (waterfront) shows the results from the EWAS. Blue lines indicate suggestive associations; red lines indicate epigenome-wide significant associations. **a** Miami plot for MATN3 (chromosome 2: 20,191,813–20,212,455). **b** Miami plot for MDGA1 (chromosome 6: 37,600,284–37,667,082). **c** Miami plot for NEP (chromosome 3: 154,741,913–154,901,518)

sites identified by our EWAS on MDGA1 levels[11]. Finally, for NEP, we identified a sole independent *trans* pQTL (rs4687657) annotated to the *ITIH4* gene (beta: 0.53; effect allele: T), as well as three *trans* genome-wide significant CpG sites (cg11645453, cg18404041 and cg06690548 annotated to *ITIH1*, *ITIH4* and *SLC7A11*, respectively). In addition to higher circulating levels of NEP, this SNP has previously been associated with lower methylation levels of cg18404041 (*ITIH4*; beta: −0.93; effect allele: T; $P = 4.20 \times 10^{-17}$) and higher methylation levels of cg11645453 (*ITIH1*; beta: 0.83; effect allele: T; $P = 1.28 \times 10^{-87}$)[12]. We performed bidirectional MR analyses to formally test whether there was a causal relationship between DNA methylation at these sites and Olink® protein levels (see methods). For each protein, MR analyses suggested that differential DNA methylation was causally associated with changes in protein levels. Conversely, altered protein levels of MATN3, MDGA1 and NEP were also causally associated with differential methylation levels at CpG sites identified by our EWAS (Supplementary Table 5).

We conducted tissue specificity and pathway enrichment analyses (KEGG and GO—see methods for details) based on genes identified by methylation for each of the 9 proteins with genome-wide significant CpG associations. Tissue-specific patterns of expression were observed for 5/9 proteins (Supplementary Data 4). Neural tissue was the most common tissue type in which genes were differentially expressed ($n = 4/5$ proteins), followed by cardiac and splenic tissue ($n = 3/5$ proteins). Gene ontology analyses revealed that genes annotated to CpG sites associated with circulating SIGLEC1 and G-CSF levels are over-represented in immune system processes, viral response and cytokine response pathways (Supplementary Data 5–6; FDR-adjusted $P$ value < 0.05). Furthermore, genes incorporating CpG sites associated with NEP levels are over-represented in metabolic pathways involving extracellular matrix components (Supplementary Data 7; FDR-adjusted $P$ value < 0.05). For CRTAM, MDGA1, MATN3, NC-Dase, SMPD1 and TN-R, there were no significant results following multiple testing correction.

**Causal evaluation of biomarkers in neurological disease.** From our GWAS, we identified a conditionally significant *cis* pQTL for plasma poliovirus receptor (PVR) levels. Furthermore, variation in the *PVR* gene has been implicated in AD[13]. Therefore, colocalisation analysis was performed to test if the same SNP variant might be driving both associations. A 200 kb region surrounding the sentinel *cis* pQTL for PVR was extracted from GWAS summary statistics for PVR levels, as well as AD[14]. Default priors were applied. There was evidence to suggest that there are two distinct causal variants for altered protein levels and AD risk within the region (PP > 0.99).

In addition to the colocalisation analysis, two-sample MR was used to test for putatively causal associations between plasma PVR levels and AD[14]. After LD pruning, only one independent SNP remained (rs7255066). Therefore, causal effect estimates were determined using the Wald ratio test, i.e., a ratio of effect per risk allele on AD to effect per risk allele on PVR levels. MR analyses indicated that PVR levels were causally associated with AD (beta = 0.17, se = 0.02, $P = 5.2 \times 10^{-10}$; Wald ratio test). Testing for horizontal pleiotropy was not possible owing to an insufficient number of instruments. Conversely, AD risk was not causally associated with PVR levels (number of SNPs: 5; Inverse variance-weighted method: beta = 0.38, se = 0.29, $P = 0.34$). The intercept from MR Egger regression was −0.08 (se: 0.08; $P = 0.42$), which does not provide strong evidence for directional pleiotropy.

**Replication of previous pQTL studies**. Replication of the pQTL findings was carried out via lookup of genotype-protein summary statistics from existing pQTL studies[4,5,15,16]. Of the 33 proteins with a conditionally significant pQTL in the present study, 15 (with 18 QTLs) were available for lookup. In total, 6/18 (33.3%) pQTLs replicated at $P < 1.25 \times 10^{-7}$ (denoting the least conservative threshold across all studies) (Supplementary Data 8). We tested the correlation of beta values for these six significant pQTLs from our study versus those reported in the literature. Notably, beta values were only available for 3/6 pQTLs in the literature. However, for these remaining 3 pQTLs, there was strong agreement between our observed values and previously reported beta statistics (rs2075803: 0.50 vs. 0.55; rs481076: 0.44 vs. 0.46 and rs1448903: 0.76 vs. 0.65, respectively). In addition, in relation to the 15 proteins from the Olink® panel which were available for look-up, we extracted beta values for all significant pQTLs associated with the levels of these proteins reported in the literature. Notably, many of these pQTLs were non-significant in our study; indeed, in this case, we wished only to determine the correlation of betas for those pQTLs reported as significant in the literature with betas from our GWAS. Beta statistics were reported for 13/15 proteins (totalling 38 pQTLs). There was a strong correlation between betas for previously reported significant pQTLs and pQTLs from our study ($r^2 = 0.89$, Supplementary Fig. 1). Finally, of the 23 pQTLs identified by FUMA which were available for look-up, 9/23 (39.1%) replicated at $P < 1.25 \times 10^{-7}$ (Supplementary Data 9).

## Discussion

Using a multi-omics approach, we identified 41 independent genome-wide significant pQTLs and 26 genome-wide significant CpG sites associated with circulating neurological protein levels. To probe the molecular mechanisms which modulate plasma protein levels, we integrated pQTL and eQTL data allowing for the examination of whether pQTLs affect gene expression. For three proteins, we found strong evidence that a common causal variant underpinned changes in transcript and protein levels. Mendelian randomisation analyses suggested that variants for two of these proteins (MDGA1 and KYNU) influence protein levels by altering gene expression. However, for one protein (DRAXIN), the converse may be true as our data suggested that altered plasma protein levels of this neurodevelopmental protein may affect gene expression, perhaps through a feedback mechanism. Genotype-protein associations for other proteins may exert their influence on protein levels through modulation of protein clearance, degradation, binding or secretion. Finally, methylation data revealed that neurological proteins were also implicated in immune, developmental and metabolic pathways.

In addition to leveraging methylation data to identify pathway enrichment for plasma proteins, identification of *trans* pQTLs may highlight previously unidentified pathways relevant to disease processes. For instance, we found that genetic variation at the inter-alpha-trypsin inhibitor heavy chain family member 4 locus (*ITIH4*) is associated with differential NEP levels (*trans* pQTL: rs4687657). In addition, two CpG sites annotated to *ITIH4* and *ITHI1* (cg18404041 and cg11645453, respectively) were associated with NEP levels. Methylation QTL analyses revealed that the SNP rs4687657 has been previously associated with lower methylation levels of cg18404041 (*ITIH4*) and higher DNA methylation levels of cg11645453 (*ITIH1*)[12]. Similarly, this SNP has been associated with higher gene expression of *ITIH4*[17] and lower protein levels of *ITIH1*[4]. Together, these data suggest that the expression of NEP, ITIH4 and ITIH1 may be co-regulated, involving inverse relationships between NEP and ITIH4 with ITIH1. Given that mutations in *NEP* have been linked to

Alzheimer's pathology and that upregulation of ITIH4 has been demonstrated in sera of AD patients[18], mechanistic studies relating to co-expression of these proteins are merited in pathological states.

In this study, a single *trans* variant (rs4857414) was associated with the circulating levels of two proteins—CD200R1 and Siglec-9. This polymorphism mapped to the *ST3GAL6-AS1* gene. ST3GAL6-AS1 is a long non-coding RNA which is associated with increased expression of ST3GAL6, an enzyme responsible for catalysing the addition of sialic acid to cell surfaces[19]. Upregulation of ST3GAL6 has been reported in multiple myeloma[20,21]; this permits evasion of immune responses against cancer cells through binding of sialic acid to Siglec receptor proteins, such as Siglec-9. The recognition of sialic acid by Siglec proteins ignites signalling cascades which promotes immune inhibitory responses[22,23]. Furthermore, CD200-CD200R interaction results in the inhibition of immune responses against multiple myeloma cells[24]. Therefore, as polymorphisms in *ST3GAL6-AS1* are associated with altered expression of Siglec-9 and CD200R, this may provide further evidence for co-regulation of these proteins in pathological milieux, such as tumorigenesis in cancers including multiple myeloma. Polymorphisms in such *trans* pQTLs may also be used to predict disease risk, progression and provide pharmacogenomic information in predicting individual patient responses to inhibition of these co-regulated proteins.

By using *cis* pQTLs as instruments for MR analyses, it is possible to test whether plasma proteins are causally associated with disease states[25]. *PVR* is a component of the AD risk-associated *APOE/TOMM40* cluster on chromosome 19 and has been hypothesised to influence risk of AD through susceptibility to viral infections[13]. However, it is unknown whether PVR is causally linked to the disease. MR analyses suggested that circulating PVR levels may be causally associated with AD and not vice versa. However, an insufficient number of instruments were available to permit testing for potential pleiotropic effects. Furthermore, colocalisation analysis revealed that independent variants in the *PVR* locus are likely causally associated with altered plasma PVR levels and AD risk. While this does not support the argument for a single causal SNP underlying both altered plasma PVR levels and AD risk, it may nevertheless suggest that genetic variation in the PVR locus is causally associated with development of AD.

The discrepancy in replication of pQTLs reported in previous studies may be due to a number of factors. First, the sample sizes of these studies ($n < 100$[15,16]; $n > 1000$[4,5]) are different from that of the present study ($n = 750$) leading to differences in statistical power. Second, diverse proteomic platforms may result in the detection of different genotype-protein associations. Our study is the first to characterise the genetic variants associated with the Olink® neurology panel and thus, the protein list and measurement technology do not overlap with platforms employed in earlier studies. Depending on platform technology, susceptibility to cross-reactive events and detection of proteins in their free, versus complexed, forms can result in inappropriate readouts. SOMAmer technology, employed in the previous pQTL studies, is a highly sensitive, aptamer-based platform which overcomes limitations associated with antibody-based methods, such as cross-reactivity[26]. Moreover, Olink® technology is particularly effective in limiting the reporting of cross-reactive events. However, when compared to these platforms, other technologies such as mass-spectrometry can produce highly accurate measurements but with low sensitivity[27]. Lack of standardisation amongst proteomic platforms, insufficient power to detect associations and differences in study demographics may all contribute to variability in the detection of pQTLs for a given protein. In addition,

we performed both FUMA (LD-based method) and COJO (stepwise conditional regression) to identify independent pQTL-protein associations and found a small overlap (17%) between SNPs identified by both methods. However, SNPs which were differentially identified by COJO and FUMA for a given protein were located within the same region. Indeed, the maximum distance between discordant SNPs for a given protein was 3 Mb.

We acknowledge several limitations in the present study. First, analyses were restricted to individuals of European descent, complicating the generalisability of our findings to individuals of other ethnic backgrounds. Second, functional enrichment analyses indicated that a number of *cis* pQTL variants may alter the amino acid sequence of the coded protein. This may lead to altered structural properties of the protein product, resulting in impaired antibody–antigen binding and consequently, the ability of assays to accurately detect protein levels. Notably, as the LBC1936 cohort consists of relatively healthy older adults, it is possible that levels of putative neurological-disease related proteins may differ in the general elderly population. Therefore, this may complicate the generalisability of our findings to other age ranges and other elderly cohorts with higher incidences of neurological and psychiatric conditions. Finally, as our findings pertain to whole blood samples, studies examining the genetic and epigenetic regulation of neurological proteins in *post-mortem* brain tissue are warranted.

In conclusion, we have identified genetic and epigenetic factors associated with neurological proteins in an older-age population. We have shown that use of a multi-omics approach can help define whether such proteins are causal in disease processes. We have shown that PVR may be causally associated with AD. Furthermore, we have provided a platform upon which future studies can interrogate pathophysiological mechanisms underlying neurological conditions. Together, this information may help inform disease biology, as well as aid in the prediction of disease risk and progression in clinical settings.

## Methods

**The Lothian Birth Cohort 1936**. The Lothian Birth Cohort 1936 (LBC1936) comprises Scottish individuals born in 1936, most of whom took part in the Scottish Mental Survey 1947 at age 11. Participants who were living within Edinburgh and the Lothians were re-contacted approximately 60 years later, 1091 consented and joined the LBC1936. Upon recruitment, participants were approximately 70 years of age (mean age: 69.6 ± 0.8 years). Participants subsequently attended four additional waves of clinical examinations every three years. Detailed genetic, epigenetic, physical, psychosocial, cognitive, health and lifestyle data are available for members of the LBC1936. Recruitment and testing of the LBC1936 cohort have been described previously[28,29]. LBC1936 participants were 49.8% female. Key inclusion/exclusion criteria for the present study are highlighted in Supplementary Fig. 2.

**Ethical approval**. Ethical permission for the LBC1936 was obtained from the Multi-Centre Research Ethics Committee for Scotland (MREC/01/0/56) and the Lothian Research Ethics Committee (LREC/2003/2/29). Written informed consent was obtained from all participants.

**Protein measurements in the Lothian Birth Cohort 1936**. Plasma was extracted from 816 blood samples collected in citrate tubes at mean age 72.5 ± 0.7 years (Wave 2). Plasma samples were analysed using a 92-plex proximity extension assay (Olink® Bioscience, Uppsala Sweden). The proteins assayed constitute the Olink® neurology biomarker panel. This panel represents proteins with established links to neuropathology, as well as exploratory proteins with roles in processes including cellular communication and immunology. In brief, 1 μL of sample was incubated in the presence of proximity antibody pairs linked to DNA reporter molecules. Upon binding of an antibody pair to their corresponding antigen, the respective DNA tails form an amplicon by proximity extension, which can be quantified by high-throughput real-time PCR. This method limits the reporting of cross-reactive events. The data were pre-processed by Olink® using NPX Manager software. Protein levels were transformed by rank-based inverse normalisation. Normalised plasma protein levels were then regressed onto age, sex, four genetic principal components of ancestry derived from the Illumina 610-Quadv1 genotype array (to control for population structure) and Olink® array plate. To obtain an estimate of

population structure, multidimensional scaling (MDS) was performed on LBC1936 genotyping data and the first four MDS components were used to control for genetic ancestry in the analytic models. Standardised residuals from these linear regression models were used in our genome-wide and epigenome-wide association studies. Pre-adjusted (raw) and transformed (rank-based inverse normalised levels) regressed on age, sex, population structure and array plate) protein levels are presented in Supplementary Data 10 and 11, respectively. The associations of pre-adjusted protein levels with biological and technical covariates are presented in Supplementary Data 12.

**Methylation preparation in the Lothian Birth Cohort 1936**. DNA from whole blood was assessed using the Illumina 450 K methylation array at the Edinburgh Clinical Research Facility (Wave 2; $n = 895$; mean age: 72.5 ± 0.7 years). Details of quality control procedures have been described in detail elsewhere[30]. Briefly, raw intensity data were background-corrected and normalised using internal controls. Following background correction, manual inspection permitted removal of low quality samples presenting issues relating to bisulphite conversion, staining signal, inadequate hybridisation or nucleotide extension. Quality control analyses were performed to remove probes with low detection rate <95% at $P < 0.01$. Samples with a low call rate (samples with <450,000 probes detected at $p$-values of less than 0.01) were also eliminated. Furthermore, samples were removed if they had a poor match between genotype and SNP control probes, or incorrect DNA methylation-predicted sex.

**Genotyping in the Lothian Birth Cohort 1936**. LBC1936 DNA samples were genotyped at the Edinburgh Clinical Research Facility using the Illumina 610-Quadv1 array (Wave 1; $n = 1005$; mean age: 69.6 ± 0.8 years; San Diego). Preparation and quality control steps have been reported previously[31]. SNPs were imputed to the 1000 G reference panel (phase 1, version 3). Individuals were excluded on the basis of sex discrepancies, relatedness, SNP call rate of less than 0.95, and evidence of non-Caucasian descent. SNPs with a call rate of greater than 0.98, minor allele frequency in excess of 0.01, and Hardy-Weinberg equilibrium test with $P \geq 0.001$ were included in analyses.

**Genome-wide association studies**. Genome-wide association analyses were conducted on 8,683,751 autosomal variants against protein residuals in 750 individuals from the Lothian Birth Cohort 1936. Linear regression was used to assess the effect of each genetic variant on the protein residuals using mach2qtl[32,33].

GWAS model: Olink® protein residuals~SNP

**Epigenome-wide association studies**. Epigenome-wide association analyses were conducted by regressing each of 459,309 CpG sites (as dependent variables) on transformed protein levels using linear regression with adjustments for age, sex, estimated white blood cell proportions (CD4+ T cells, CD8+ T cells, B cells, Natural Killer Cells and granulocytes) and technical covariates (plate, position, array, hybridisation, date). White blood cell proportions were estimated from methylation data using the Houseman method[34]. Outliers for white blood cell proportions ($n = 22$) were excluded prior to analyses. Complete methylation and proteomic data were available for 692 individuals. Genome-wide significant CpG associations mapping to sites with underlying polymorphisms were excluded, as well as those predicted to cross-hybridise based on findings by Chen et al.[35]. Analyses were performed using the limma package in R[36].

EWAS model: CpG site~Olink® protein residuals + age + sex + estimated white blood cell proportions + array + plate + date + set + position

Pathway enrichment was assessed among KEGG pathways and Gene Ontology (GO) terms via hypergeometric tests using the *phyper* function in R. All gene symbols from the 450 K array annotation (null set of sites) were converted to Entrez IDs using biomaRt[37,38]. GO terms and their corresponding gene sets were obtained from the Molecular Signatures Database (MSigDB)-C5[39] while KEGG pathways were downloaded from the KEGG REST server[40]. Furthermore, tissue specificity analyses were conducted using the GENE2FUNC function in FUnctional Mapping and Annotation (FUMA). Differentially expressed gene sets with Bonferroni-corrected $P$ values of <0.05 and an absolute log-fold change of ≥0.58 (default settings) were considered to be enriched in a given tissue type (GTEx v7).

**OSCA**. We also performed EWAS analyses of Olink® protein levels using OmicS-data-based Complex trait Analysis software (OSCA). We carried out OSCA as an additional EWAS analysis as it has recently been shown to identify less spurious associations when compared to other methods (including linear regression)[41]. CpG site was the independent variable whereas Olink® protein levels were input as dependent variables. Models were adjusted for age, sex, estimated white blood cell proportions (CD4+ T cells, CD8+ T cells, B cells, Natural Killer Cells and granulocytes) and technical covariates (plate, position, array, hybridisation, date) as in the previous section. The MOMENT method was used to test for associations between traits of interest and DNAm at individual probes. MOMENT is a mixed-linear-model-based method that can account for unobserved confounders and the correlation between distal probes which may be introduced by such confounders.

**Conditional and joint analysis**. We performed approximate genome-wide step-wise conditional analysis through GCTA-COJO using the 'cojo-slct' option as the primary means to identify independent genetic-protein associations[42]. Individual level genotype data were used with default settings of the software.

**Functional mapping and annotation of pQTLs**. In addition to GCTA-COJO, the identification of independent pQTL variants from the GWAS which yielded significant genotype-protein associations, and their subsequent functional annotation, were performed using the independent SNP algorithm implemented in FUMA analysis[43]. Initial independent significant SNPs were identified using the SNP2GENE function. These were defined as variants with a $P$ value of $<5 \times 10^{-8}$ that were independent of other genome-wide significant SNPs at $r^2 < 0.6$. Lead independent SNPs were further defined as the initial independent significant SNPs that were independent from each other at $r^2 < 0.1$. Independent significant SNPs were functionally annotated using ANNOVAR[44] and Ensembl genes (build 85).

**Characterisation of *cis* and *trans* effects**. Genome-wide significant pQTLs and CpG sites were categorised into *cis* and *trans* effects. *Cis* associations were defined as loci which reside within 10 Mb of the TSS of the gene encoding the protein of interest. *Trans* effects were defined as those loci which lay outside of this region or were located on a chromosome distinct from that which harboured the gene TSS. TSS positions were defined using the biomaRt package in R[37,38] and Ensembl v83.

**Identification of overlap between *cis* pQTLs and eQTLs**. We cross-referenced sentinel *cis* pQTLs with publicly available *cis* eQTL data from the eQTLGen consortium[45]. *Cis* eQTLs were filtered to retain only variants with $P < 5.4 \times 10^{-10}$. Furthermore, only *cis* eQTLs for the same gene as the *cis* pQTL protein were retained. These associations were then tested for colocalisation.

**Colocalisation analysis**. To test the hypothesis that a single causal variant might underlie both an eQTL and pQTL, resulting in modulation of transcript and protein levels, we conducted Bayesian tests of colocalisation. Colocalisation analyses were performed using the coloc package in R[46]. For each pQTL variant, a 200 kb region (upstream and downstream) was extracted from our GWAS summary statistics for each protein of interest. This window previously has been recommended in order to capture *cis* eQTLs, which often lie within 100 kb of their target gene[47]. Expression QTLs for genes within this region were extracted from eQTLGen consortium summary statistics and subset to the gene encoding the protein of interest[45]. All SNPs shared by transcripts and proteins were used to determine the posterior probability for five distinct hypothesis. Default priors were applied. Posterior probabilities (PP) > 0.95 provided strong evidence in favour of a given hypothesis. Hypothesis 4 states that two association signals were attributable to the same causal variant. Associations with PP4 > 0.95 were deemed highly likely to colocalise. Associations with PP3 > 0.95 provided strong evidence for hypothesis 3 that there were independent causal variants for protein levels and gene expression. In this study, hypothesis 2 referred to a causal variant for condition 2 (gene expression only) whereas hypothesis 1 represented a causal variant for protein levels only. Associations with PP0 > 0.95 (for hypothesis 0) indicated that it is highly likely there were no causal variants for either trait in the region.

**Mendelian randomisation**. Two-sample bidirectional Mendelian randomisation was used to test for putatively causal relationships between (i) PVR, a cell-surface glycoprotein, and AD risk, (ii) gene expression and plasma protein levels and (iii) DNA methylation and plasma protein levels. Pruned variants (LD $r^2 < 0.1$) were used as instrumental variables (IV) in MR analyses. In cases where only one independent SNP remained after LD pruning, causal effect estimates were determined using the Wald ratio test. When multiple independent variants were present, and if no evidence of directional pleiotropy was present (non-significant MR-Egger intercept), multi-SNP MR was conducted using inverse variance-weighted estimates. All MR analyses were conducted using MRbase[48].

(i) While 72 genome-wide significant *cis* pQTLs were identified for PVR levels, only one SNP (rs7255066) remained after LD pruning. Five independent SNPs were identified and used as IV to test for a causal relationship between AD risk and altered plasma PVR levels.

(ii) Expression QTLs obtained from eQTLGen consortium were used as IV to test whether changes in gene expression were causally associated with protein levels[45]. Protein QTLs identified by our GWAS were used as IV to test whether protein levels were causally associated with altered gene expression.

(iii) For the three proteins with GWAS and EWAS associations (MATN3, MDGA1 and NEP), we wished to test whether methylation affected protein levels and/or whether protein levels affected methylation. We queried Phenoscanner to examine whether pQTLs for protein levels of MATN3, MDGA1 and NEP, identified in this study, were previously identified as methylation QTLs (mQTLs) for corresponding genome-wide significant CpG sites[49]. Methylation QTLs were used as IV to test whether changes in DNA methylation were causally associated with Olink® protein levels. Conversely, pQTLs were used as IV to determine whether altered protein

levels were causally linked to differential methylation levels. Of note, as methylation of the 11 *cis* CpG sites associated with differential MDGA1 levels in our study are highly inter-correlated (Supplementary Fig. 3), we considered only the most significant *cis* CpG site (cg20053110) for MR analyses.

**Reporting summary**. Further information on research design is available in the Nature Research Reporting Summary linked to this article.

## Data availability
Full and openly accessible summary statistics from the association studies on Olink® neurology protein levels are available on the University of Edinburgh Datashare site (https://datashare.is.ed.ac.uk/). For GWAS data, see: https://datashare.is.ed.ac.uk/handle/10283/3366; https://doi.org/10.7488/ds/2580. For EWAS data, see: https://datashare.is.ed.ac.uk/handle/10283/3367; https://doi.org/10.7488/ds/2581.

## Code availability
Code will be available from the authors on request.

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

## Acknowledgements

The authors thank all LBC1936 study participants and research team members who have contributed, and continue to contribute, to ongoing LBC1936 studies. The LBC1936 is supported by Age UK (Disconnected Mind program) and the Medical Research Council (MR/M01311/1). Methylation typing was supported by Centre for Cognitive Ageing and Cognitive Epidemiology (Pilot Fund award), Age UK, The Wellcome Trust Institutional Strategic Support Fund, The University of Edinburgh, and The University of Queensland. This work was conducted in the Centre for Cognitive Ageing and Cognitive Epidemiology, which is supported by the Medical Research Council and Biotechnology and Biological Sciences Research Council (MR/K026992/1), and which supports I.J.D. We acknowledge NIH Grants R01AG054628 and R01AG05462802S1 for supporting this research and Grant P2CHD042849 for supporting the Population Research Center at the University of Texas. R.F.H. and A.J.S. are supported by funding from the Wellcome Trust 4-year PhD in Translational Neuroscience–training the next generation of basic neuroscientists to embrace clinical research [108890/Z/15/Z]. A.S. is supported by a Medical Research Council PhD Studentship in Precision Medicine with funding by the Medical Research Council Doctoral Training Programme and the University of Edinburgh College of Medicine and Veterinary Medicine. D.L.Mc.C. and R.E.M. are supported by Alzheimer's Research UK major project grant ARUK-PG2017B−10. This research was supported by Australian National Health and Medical Research Council (grants 1010374, 1046880 and 1113400) and by the Australian Research Council (DP160102400). P.M.V., N.R.W. and A.F.M. are supported by the NHMRC Fellowship Scheme (1078037, 1078901 and 1083656).

## Author contributions

Conception and design: R.F.H. and R.E.M. Data analysis: R.F.H., D.L.Mc.C., D.C.L. and R.E.M. Drafting the article: R.F.H. and R.E.M. Revision of the article: all authors.

## Additional information

**Competing interests:** The authors declare no competing interests.

