## [Peer Review File · Nature Communications]

Reviewers' Comments:

Reviewer #1:

Remarks to the Author:

The authors report on a genome-wide association study (GWAS) and epigenome-wide association study (epiGWAS) on the levels of 92 proteins represented in the Olink® neurology biomarker panel in an elderly population of subjects (N=750) from the Lothian birth cohort 1936. They report on 62 independent genome-wide significant loci for 37 proteins and 68 epigenome-wide significant sites associated with the levels of 7 proteins. Integrating pQTL and eQTL data they found evidence for 5 of the circulating proteins, that a common causal variant underpinned changes in transcription and protein levels. Applying a two-sample Mendelian Randomisation approach they further provide some evidence for a causal association between circulating levels of poliovirus receptor (PVR) and Alzheimer's disease (AD).

I had difficulties extracting a major message in terms of a specific finding that would provide in depth mechanistic insights or provide a convincing starting point for further investigation. In my view the results can be seen as a valuable resource alongside other data sources on pQTLs (e.g. Suhre et al. 2017; Sun et al. 2018), eQTLs, or meQTLs, some of which took a much broader approach (i.e. covering much more proteins (e.g. Sun et al 2018) and also included much more individuals (eg. Sun et al. 2018; for details see below).

Major comments:

Probably the most interesting finding is the reported association between poliovirus receptor PVR and risk of Alzheimer's disease. However, there are some concerns with this result: First, the observation of a pQTL cis association originated from GWAS where the pQTL for PVR was just one among many other pQTLs, each of which – following the logic of the overall approach – will probably have been checked for possible associations with neurological or psychiatric phenotypes. This in a way raises the issue of multiple hypothesis testing; Second, the authors found two distinct variants for altered protein levels and risk of AD, which doesn't strengthen the argument for a causal relationship; Third, the two-sample Mendelian Randomization study relies on a single variant, which precluded testing for horizontal pleiotropy as is also acknowledged by the authors. I think the conclusions regarding the relationship of PVR and risk of Alzheimer's disease should be tuned down and probably even be removed from the Abstract unless the authors can further solidify their hypothesis by providing additional data.

The number of circulating proteins assessed in this study (N=92) is relatively small compared to recent studies (N=1.124 in Suhre et al. 2017; N=2.994 in Sun et al. 2018) as is the overall sample size (N=750 individuals in the current study compared to N=3.301 individuals in Sun et al. 2018) and there is no replication of significant pQTLs in a second dataset. While this can be justified in a screening effort the low proportion (39%) of replicated pQTLs reported in previous studies is a concern. The authors provide a number of potential explanations for this but overall this doesn't raise confidence in the robustness of the available pQTL data for circulating proteins.

From a technical standpoint I wonder why the authors used FUMA as the primary analysis and GCTA-COJO as the secondary analysis. According to my understanding GCTA-COJO is more powerful than FUMA and should thus be defined as the primary analysis with FUMA being the secondary analysis. Also, from looking at the results for the COJO analysis it is not clear to which extent the 35 proteins for which the authors identified significant pQTLs overlapped with the 37 proteins for which the authors identified independent pQTLs by FUMA. This should be specified in the text.

The probability cut-off for the posterior probability (PP) of 0.75 seems too low as it bears some risk of reporting on false positive associations. I would recommend applying a PP of >0.95. Results were obtained in an elderly population of subjects, which is not described in the manuscript unless I overlooked something. I wonder whether the authors could include a supplementary table providing key characteristics of their study sample including information on

key inclusion / exclusion criteria and the prevalence of major (neurological and psychiatric) diseases (if not an exclusion criterion anyway) I further wonder whether this could be considered a survivor cohort of rather overly healthy subjects and how this might have influenced the results (please discuss)

The authors found neural tissue to be the most common tissue type in which genes were differentially expressed. Isn't this very expected given the nature of the (Neuro)panel. In other words, was there a significant enrichment of any particular KEGG or GO category for the 7 proteins compared to the remaining proteins represented by their panel?

Minor:

Introduction: the meaning of "genetic architecture" is somewhat vague and I am not sure this is an appropriate term. What exactly do the authors mean with "genetic architecture of neurology-related proteins"?

Figures: I think the authors mixed up figures 1b and 1c. For instance the statement in line 111-113 seems to refer to Figure 1c rather than Figure 1b whereas the statement in Line 115 refers to figure 1b rather than 1c.

Tables: in Line 118-120 the authors refer to Supplementary Tabel 2 when speaking about two specific SNPs. However, I didn't find these SNPs in the table.

Methods: I wonder whether the authors could provide some addition information on the proteins captured by the Olink platform.

Line 142-143: The following statement is not entirely clear and requires rephrasing of some additional explanation: "Finally, for CLM-6, there was strong evidence ($PP > 0.75$) for a causal variant for gene expression only in the locus"

Line 275: "long-coding RNA" I assume the authors intended to write "long non-coding RNA"

Line 327: the authors write that "...functional enrichment analyses indicated that a number of cisQTL variants alter the amino acid sequence of the coded protein which may impact the quantitative protein assay." Could they expand a bit on explaining this interpretation?

Reviewer #2:

Remarks to the Author:

The manuscript by Hillary et al entitled, "Genetic and epigenetic architectures of neurological protein biomarkers in the Lothian Birth Cohort", sought to identify genetic and epigenetic loci related to inter-individual differences in circulating serum proteins relevant to neurologic processes and diseases. The authors identified 62 independent genetic variants that explain a large proportion of inter-individual variance (~5-53%) in 37 circulating serum proteins. In addition, DNA methylation levels at 68 loci showed significant associations with some circulating neurological protein levels. Mendelian Randomization techniques were employed to determine whether pQTLs influenced protein levels through gene expression and for 4 proteins the data suggests a causal pathway through gene expression. Finally, the authors provide results that suggest a causal association between Alzheimers Disease and one of the circulating proteins – poliovirus receptor.

This is an important area of research with potential high impact by advancing our knowledge of genetic variants that control neurologically-relevant proteins and the utility of accessible tissues when studying brain-based disorders. The data and results of this manuscript are also likely to provide a valuable resource to the scientific community. However, I have a few major concerns that should be addressed and several other suggestions intended to strengthen the manuscript.

Results Section

1. The pre- and post-transformed and adjusted protein level distributions should be shown and/or

a descriptive table of the study sample and whether the protein levels are related to biological and technical covariates.

2. The authors use all discovery SNPs with a $P < 1$ to generate their PRS. They find no association with the brain protein biomarkers using this threshold that includes all SNPs, although, depression showed the strongest negative correlation. Both simulation and empirical analyses (e.g. PMID: 30104760) has shown depression is the most highly polygenic disorder which taken with the authors results and inclusion of all SNPs in the genome suggests different PRS liability and thresholds could impact their results. How are the PRS-protein results impacted when different discovery thresholds are used (e.g. $P < 0.5$, $P < 0.25$, $P < 0.1$, $P < 0.01$)?

3. The authors state that they adjusted for cell composition in their EWAS analysis using basophils, eosinophils, neutrophils, lymphocytes, and monocytes cell count information. I am concerned that one of the main results, "The majority of the protein-CpG associations were attributable to CRTAM ... which is upregulated in CD4+ and CD8+ cells" is driven by differences in these cell types that were not captured in the cell type adjustment. The authors should use the Houseman method (PMID: 22568884) to specifically compute the proportion of CD4+ and CD8+ cell types and include them as covariates in their analytic models.

4. When testing for colocalization of SNPs driving multiple associations, the authors state they selected a "200kb region surrounding the sentinel cis pQTL". Because physical distance does not capture LD structure, the authors should instead consider selection of SNPs to test for colocalization based on LD block as opposed to absolute physical distance.

5. One of the major innovative features of the study that is highlighted by the authors is the integration of data across omics types. However, much of this type of analysis is carried out using expression databases as opposed to testing DNA Methylation Outcome relationships using data they collected from the same study participants. Am I missing the rationale for why DNA methylation could be mediating the genetic-protein level/disease associations? Could you perform statistical analyses to see if methylation is mediating the genetic protein level associations? work is linking

6. It is unclear on line 148, which pruned cis QTLs were instrumental variables? Was it pQTLs, eQTLs?

Methods

7. It is unclear how many samples and SNPs were filtered out at each quality control step or how many samples and loci were tested in each analysis. It would also be beneficial to understand the number of individuals that contributed to both the GWAS and EWAS analyses. Please provide a flow chart and/or report quality control results in the results/methods section.

8. The authors state that sex, age, and genetic ancestry were adjusted for at the protein measurement level and the residuals were used in downstream EWAS and GWAS (described in section 4.2). What is the rationale for adjusting for genetic ancestry at the protein level as opposed to in the GWAS statistical models? In section 4.7, the epigenome analyses, sex and age appear to be included in the EWAS models as a covariate, it isn't clear to me why these would be adjusted for again?

9. The methods section describing GWAS imputation seems to fit better further up near the description of the genotyping measurements and before the polygenic risk score description.

10. There is no description of how genetic ancestry was ascertained

11. The description of GO enrichment tests does not specify what the null set of sites included. For example, was it all loci in the genome, all loci measured on the 450K, etc? It should be based on the set of measured/assayed genomic loci/genes. Also, a description of how CpG sites were mapped to genes for GO testing is needed.

12. Statistical models used for GWAS and EWAS should be specified

13. There is no description of the Mendelian Randomization analyses in the method section

14. The authors seem to have used some outdated summary statistics for PRS or did not specify the exact source of the summary statistics (e.g. autism)

Response to Editors and Reviewers

We are very grateful for the comments provided by the editor and each of the external reviewers of this manuscript. The comments are encouraging and we hope that you will agree that the updated manuscript has satisfactorily addressed these comments and concerns, and that the overall manuscript is strengthened. We agree that these results can provide a useful molecular database of protein-genetic and protein-epigenetic associations upon which other researchers may use as a starting point for investigations pertinent to causal relationships between plasma proteins and disease. Furthermore, a broad range of researchers may use this resource in relation to mechanistic interrogations of molecular mechanisms which underpin various disease states. Please see below, in blue, a detailed response to comments and concerns raised by the editor (concerning data availability) and the reviewers. Line numbers refers to the manuscript file with highlighted changes.

Response to the reviewers

Reviewer #1

Major comments

Comment 1.1

“Probably the most interesting finding is the reported association between poliovirus receptor PVR and risk of Alzheimer’s disease. However, there are some concerns with this result: First, the observation of a pQTL cis association originated from GWAS where the pQTL for PVR was just one among many other pQTLs, each of which – following the logic of the overall approach – will probably have been checked for possible associations with neurological or psychiatric phenotypes. This in a way raises the issue of multiple hypothesis testing.

We thank the reviewer for this comment and appreciate these concerns relating to interpretation of this interesting finding. Firstly, in relation to concerns surrounding multiple testing correction, we tested only for an association between poliovirus receptor (PVR) and Alzheimer’s disease (AD) risk. We did not test any other proteins or any other disease for the MR analysis. All significant *cis* pQTLs identified by our GWAS on PVR levels (n = 72) were used as input for the Mendelian randomisation analysis to test whether PVR levels were causally associated with AD risk. However, following linkage disequilibrium-based clumping, only one available variant remained to be used as the instrumental variable in the analysis leaving just one independent test. To clarify this, and in response to reviewer #2, we have added in a methods section for the Mendelian randomisation analyses (section 4.14):

“Two-sample bidirectional Mendelian Randomisation was used to test for putatively causal relationships between (i) PVR, a cell-surface glycoprotein, and AD risk, (ii) gene expression and plasma protein levels and (iii) DNA methylation and plasma protein levels. Pruned variants ($LD r^2 < 0.1$) were used as instrumental variables (IV) in MR analyses. In cases where only one independent SNP remained after LD pruning, causal effect estimates were determined using the Wald ratio test. When multiple independent variants were present, and if no evidence of directional pleiotropy was present (non-significant MR-Egger intercept), multi-SNP MR was conducted using inverse variance-weighted estimates. All MR analyses were conducted using MRbase (Hemani et al., 2018).

- (i) *Either pQTLs for PVR levels identified in this study or genome-wide significant SNPs from a recent GWAS on AD risk were used as the appropriate IV in the bidirectional MR analysis (Jansen et al., 2019). While 72 genome-wide significant cis pQTLs were identified for PVR levels, only one SNP (rs7255066) remained after LD pruning. Five independent SNPs were identified and used as IV to test for a causal relationship between AD risk and altered plasma PVR levels”.*

Second, the authors found two distinct variants for altered protein levels and risk of AD, which doesn't strengthen the argument for a causal relationship; Third, the two-sample Mendelian Randomization study relies on a single variant, which precluded testing for horizontal pleiotropy as is also acknowledged by the authors. I think the conclusions regarding the relationship of PVR and risk of Alzheimer's disease should be tuned down and probably even be removed from the Abstract unless the authors can further solidify their hypothesis by providing additional data.”

Regarding the second and third points concerning the strength of the results showing a causal association between PVR and AD, we agree that a more cautious interpretation should be taken. As a result, we have removed this result from the abstract and in the discussion section, have made the following changes to make our conclusions more conservative:

1. The final line of paragraph 1 of the discussion section has been removed:

“Finally, we found evidence of a causal association between poliovirus receptor protein and AD”

2. The following text has been removed from paragraph 4 of the discussion section:

“One possible explanation for this is that variation at a pQTL is associated with circulating PVR levels but may not reflect disease-relevant mechanisms, such as altered PVR expression in neural tissue. Indeed, genetic variation at a distinct site in the locus may directly influence AD risk through a distinct neuropathological mechanism leading to development of the disease. Further studies are merited to define possible pleiotropic

effects, as well as the precise relationship between genetic variation in the PVR locus and susceptibility to AD to refine its potential as a biomarker for the disease.”

3. The word “Importantly” has been removed from line 369 in the closing paragraph of the conclusion section.

We have also included the following additional points of discussion on lines 327-331 to further apply a cautious interpretation of these findings:

“Furthermore, colocalisation analysis revealed that independent variants in the PVR locus are likely causally associated with altered plasma PVR levels and AD risk. While this does not support the argument for a single causal SNP underlying both altered plasma PVR levels and AD risk, it may nevertheless suggest that variation in the PVR locus is causally associated with development of AD.”

Comment 1.2

“The number of circulating proteins assessed in this study (N=92) is relatively small compared to recent studies (N=1.124 in Suhre et al. 2017; N=2.994 in Sun et al. 2018) as is the overall sample size (N=750 individuals in the current study compared to N=3.301 individuals in Sun et al. 2018) and there is no replication of significant pQTLs in a second dataset. While this can be justified in a screening effort the low proportion (39%) of replicated pQTLs reported in previous studies is a concern. The authors provide a number of potential explanations for this but overall this doesn’t raise confidence in the robustness of the available pQTL data for circulating proteins.”

We appreciate this concern regarding the robustness of available pQTL data for circulating proteins. Indeed, we found that 39% of independent pQTLs identified by FUMA and 33% of pQTLs identified by conditional and joint analysis in our study replicated variants reported by earlier pQTL studies. We hope it is appreciated that our finding reflects that caution should be exercised in the field and highlight that different pQTLs within the same genomic region may be identified using different methodologies. We acknowledge that our sample size is smaller than certain studies (Suhre *et al.*, 2017, Sun *et al.*, 2018), but also larger than other examined earlier pQTL studies (Lourdusamy *et al.*, 2012, Di Narzo *et al.*, 2017) which we highlight on Line 333. However, while our sample size is not the largest of pQTL studies to date, it is the only such pQTL study to incorporate methylation data which has allowed us to further inform the molecular mechanisms underpinning inter-individual variation in plasma protein levels. To further clarify the disparity between proteomic platforms employed in these studies, and that ours is the first to report pQTLs of proteins assayed in the Olink® neurology panel, we have added the following text in lines 337-339:

“Our study is the first to characterise the genetic variants associated with the Olink® neurology panel and thus, the protein list and measurement technology do not overlap with platforms employed in earlier studies”

Additionally, we have carried out further analyses to determine the agreement between beta values firstly for pQTLs which were replicated in our study with corresponding betas in the literature (n = 3 SNPs with available beta values). We also wished to examine whether beta statistics for pQTLs reported as significantly associated with protein levels in previous studies were correlated with beta values from our GWAS. Of note, these SNPs were not all significant in our study, rather we were aiming to determine whether there was a high correlation between betas for significant pQTLs reported in summary statistics for previous studies and our beta values (n = 34 SNPs with available beta values; $r^2 = 0.89$; Supplementary Data 2). The following text has been added to lines 260-273 to highlight this new analyses:

“We tested the correlation of beta values for these six significant pQTLs from our study versus those reported in the literature. Notably, beta values were only available for 3/6 pQTLs in the literature. However, for these remaining 3 pQTLs, there was strong agreement between our observed values and previously reported beta statistics (rs2075803: 0.50 vs 0.55; rs481076: 0.44 vs 0.46 and rs1448903: 0.76 vs 0.65, respectively). Additionally, in relation to the 15 proteins from the Olink panel which were available for look-up, we extracted beta values for all significant pQTLs associated with the levels of these proteins reported in the literature. Notably, many of these pQTLs were non-significant in our study; indeed, in this case, we wished only to determine the correlation of betas for those pQTLs reported as significant in the literature with betas from our GWAS. Beta statistics were reported for 13/15 proteins (totalling 38 pQTLs). There was a strong correlation between betas for previously reported significant pQTLs and pQTLs from our study ($r^2 = 0.89$, Supplementary Data 2). Finally, of the 23 pQTLs identified by FUMA which were available for look-up, 9/23 (39.1%) replicated at $P < 1.25 \times 10^{-7}$ (Supplementary Table 14).”

Comment 1.3

“From a technical standpoint I wonder why the authors used FUMA as the primary analysis and GCTA-COJO as the secondary analysis. According to my understanding GCTA-COJO is more powerful than FUMA and should thus be defined as the primary analysis with FUMA being the secondary analysis. Also, from looking at the results for the COJO analysis it is not clear to which extent the 35 proteins for which the authors identified significant pQTLs overlapped with the 37 proteins for which the authors identified independent pQTLs by FUMA. This should be specified in the text.”

We thank the reviewer for raising this concern and as a result, GCTA-COJO has been employed as the primary analysis with FUMA now denoted as the secondary analysis. The results in sections: Results (2.1, 2.2 and 2.6) have been re-written to reflect this substantial change in primary analysis. Figures 1 and 2 have also been updated to reflect this re-analysis. Furthermore, in Supplementary Table 2 (showing conditionally significant pQTLs), SNPs identified by both FUMA and COJO have been emboldened for the reader's benefit. Additionally, we have calculated measures of LD (r^2 ; per protein)

for SNPs which were differentially identified by COJO and FUMA (Supplementary Table 4). Indeed, to clarify the specific overlap between FUMA and COJO, we have added the following text from lines 131-146 to the manuscript:

“We also used an alternative method, FUMA (FUnctional Mapping and Annotation) to find independent pQTLs. This approach identified 62 significant pQTLs associated with the levels of 37 proteins (90.3% cis and 9.7% trans effects; Bonferroni-corrected level of significance: $P < 5.4 \times 10^{-10}$) (Supplementary Table 3). In contrast to GCTA-COJO, FUMA retains the most significant pQTL to identify independent signals through linkage disequilibrium (LD)-based pruning; therefore, variants were identified for all 37 proteins. Seven independent pQTLs associated with the levels of 6 proteins were found using both approaches whereas the remaining SNPs identified by COJO for a given protein were located within the same locus as corresponding SNPs identified by FUMA (overlapping SNPs highlighted in Supplementary Table 2). Additionally, we calculated a measure of LD (r^2) between SNPs which were discordant between COJO and FUMA. As 7 independent pQTLs were identified by both methods, this left 34 (41 – 7) discordant SNPs from COJO and 55 (62 – 7) discordant SNPs from FUMA. Furthermore, as some proteins contained multiple QTLs this resulted in 74 SNP-SNP comparisons between COJO and FUMA. SNPs which exhibited an r^2 coefficient > 0.75 were considered to show evidence of replication (through LD) between both methods. In total, 27 COJO SNP-FUMA SNP comparisons exhibited an $r^2 > 0.75$. This consisted of 26 unique SNPs identified by COJO and encompassed 24 proteins (Supplementary Table 4).”

Comment 1.4

“The probability cut-off for the posterior probability (PP) of 0.75 seems too low as it bears some risk of reporting on false positive associations. I would recommend applying a PP of >0.95 .”

We thank the reviewer for this recommendation. The probability cut-off for colocalisation analyses has been raised to > 0.95 and has been applied to the new eQTL-pQTL associations identified following COJO analyses. As a result, the text in section 2.2 and Supplementary Table 5 have been updated to reflect these changes.

Comment 1.5

“Results were obtained in an elderly population of subjects, which is not described in the manuscript unless I overlooked something. I wonder whether the authors could include a supplementary table providing key characteristics of their study sample including information on key inclusion / exclusion criteria and the prevalence of major (neurological and psychiatric) diseases (if not an exclusion criterion anyway). I further wonder whether this could be considered a survivor cohort of rather overly healthy subjects and how this

might have influenced the results (please discuss)”

To address this comment, we aim to clarify that participants of the Lothian Birth Cohort 1936 study (mean age: 70 at recruitment, mean age: 73 for proteomic analyses), represent a cohort of healthy older adults. Firstly, in conjunction with comments from reviewer #2, we have added a flow chart of inclusion/exclusion criteria for GWAS/EWAS studies and explicitly state that for inclusion, participants must not have had a neurodegenerative disease at Wave 1 (Supplementary Data 3). We also have included the following text in the introduction and discussion to further address that this cohort aims to examine ageing in healthy elderly adults and critique how this may affect generalisability of the presented findings:

Lines 84-87 in the Introduction section has been updated as follows:

“Here, genome- and epigenome-wide association studies (GWAS/EWAS) are carried out on the plasma levels of 92 neurological proteins in 750 ~~participants~~ healthy older adults from the Lothian Birth Cohort 1936 study (mean age: 73; levels adjusted for age, sex, population structure and array plate; hereafter simply referred to as protein levels).”

Lines 360-364 in the Discussion section have also been added:

“Notably, as the LBC1936 cohort consists of relatively healthy older adults, it is possible that levels of putative neurological-disease related proteins may differ in the general elderly population. Therefore, this may complicate the generalisability of our findings to other age ranges and other elderly cohorts with higher incidences of neurological and psychiatric conditions.”

Comment 1.6

“The authors found neural tissue to be the most common tissue type in which genes were differentially expressed. Isn’t this very expected given the nature of the (Neuro)panel. In other words, was there a significant enrichment of any particular KEGG or GO category for the 7 proteins compared to the remaining proteins represented by their panel?”

In this portion of our analyses, we would like to clarify that we only performed pathway enrichment analyses for those proteins (formerly $n = 7$; following EWAS adjustment: $n = 9$) which exhibited genome-wide significant hits. For example, MDGA1 harboured 11 genome-wide CpG associations (at $P < 3.9 \times 10^{-10}$). We then took all genes annotated to CpG sites significantly associated with MDGA1 levels at $P < 1 \times 10^{-5}$ ($n = 9$ genes) and tested whether this gene set were enriched for pathways when compared to the null set of sites (all genes on 450k array). This was done in an effort to highlight, for each individual protein, possible biological pathways in which these proteins are implicated. Therefore, the presented results in Supplementary Tables 9-11 related only to these 9 proteins and do not bear relevance the remaining 83 proteins which lack significant CpG

site associations. Following tissue enrichment analysis, we found that gene sets associated with 5/9 proteins were differentially expressed in certain tissue types. Notably, the most common tissue type in which these gene sets associated with these proteins (4/5 proteins) were differentially expressed was neural tissue. However, we also show that the majority of these proteins are also differentially expressed in cardiac and splenic tissue which may be of biological importance and interest to others wishing to examine proteins involved in this panel.

Minor comments

Comment 1.7

"The meaning of "genetic architecture" is somewhat vague and I am not sure this is an appropriate term. What exactly do the authors mean with "genetic architecture of neurology-related proteins"?"

We accept that this definition, although previously used to describe pQTL studies, may be considered vague. As a result, the title of the manuscript has been updated to:

"Genome and epigenome wide studies of neurological protein biomarkers in the Lothian Birth Cohort 1936"

Additionally, subtitles for the results sections 2.1 and 2.3 have been changed from:

"2.1 (2.3) Genetic (Epigenetic) architecture of neurological protein biomarkers"

to

"2.1 (2.3) Genome (Epigenome) wide study of neurological protein biomarkers"

Furthermore, incidences of the term "genetic architecture" have been removed from the manuscript.

Comment 1.8

"I think the authors mixed up figures 1b and 1c. For instance the statement in line 111-113 seems to refer to Figure 1c rather than Figure 1b whereas the statement in Line 115 refers to figure 1b rather than 1c. In Line 118-120 the authors refer to Supplementary Table 2 when speaking about two specific SNPs. However, I didn't find these SNPs in the table."

We thank the reviewer for noticing the erroneously named figure titles, which have now been corrected to reflect the respective figures. Additionally, in what is now Supplementary Table 3, we ensure that these two SNPs are present in the table. However, following implementing GCTA-COJO as the primary analysis, reference to these SNPs has removed from the text.

Comment 1.9

"I wonder whether the authors could provide some addition information on the proteins captured by the Olink platform."

To address this comment, the following text has been added on lines 392-393 in methods section 4.2:

"This panel represents proteins with established links to neuropathology as well as exploratory proteins with roles in processes including cellular communication and immunology."

Comment 1.10

"The following statement is not entirely clear and requires rephrasing of some additional explanation: "Finally, for CLM-6, there was strong evidence (PP> 0.75) for a causal variant for gene expression only in the locus"'"

To address this comment, we have amended the text to clarify that following colocalisation analyses, there was weak evidence for hypothesis 2 which states that there is a causal variant for gene expression of CLM-6, but not protein levels, within the respective locus:

"Finally, for CLM-6, there was weak evidence (PP > 0.75) for a causal variant affecting gene expression, but not protein levels, within the locus"

Comment 1.11

"Line 275: "long-coding RNA" I assume the authors intended to write "long non-coding RNA"

We thank the reviewer for noticing this error which has been corrected to "long non-coding RNA" on Line 306.

Comment 1.12

"The authors write that "...functional enrichment analyses indicated that a number of cisQTL variants alter the amino acid sequence of the coded protein which may impact the quantitative protein assay." Could they expand a bit on explaining this interpretation?"

To clarify our interpretation we have updated the text on lines 357-360 to:

"Secondly, functional enrichment analyses indicated that a number of cis pQTL variants may alter the amino acid sequence of the coded protein. This may lead to altered structural properties of the protein product, resulting in impaired antibody-antigen binding and consequently, the ability of assays to accurately detect protein levels."

Reviewer #2

Major comments

Comment 2.1

"The pre- and post-transformed and adjusted protein level distributions should be shown and/or a descriptive table of the study sample and whether the protein levels are related to biological and technical covariates."

Pre-adjusted (raw) and transformed (rank-based inverse normalised levels regressed on age, sex, population structure and array plate) protein level distributions are now presented in Supplementary Data 4 and 5, respectively. The associations of pre-adjusted protein levels with biological and technical covariates (age, sex, four genetic principal components of ancestry and Olink® array plate) are presented in Supplementary Table 15. This has been highlighted in the text on lines 405-408:

"Pre-adjusted (raw) and transformed (rank-based inverse normalised levels regressed on age, sex, population structure and array plate) protein levels are presented in Supplementary Data 4 and 5, respectively. The associations of pre-adjusted protein levels with biological and technical covariates are presented in Supplementary Table 15."

Furthermore, the following text has been added to lines 385-387:

"LBC1936 participants were 49.8% female. Key inclusion/exclusion criteria for the present study are highlighted in Supplementary Data 3."

Comment 2.2

"The authors use all discovery SNPs with a $P < 1$ to generate their PRS. They find no

association with the brain protein biomarkers using this threshold that includes all SNPs, although, depression showed the strongest negative correlation. Both simulation and empirical analyses (e.g. PMID: 30104760) has shown depression is the most highly polygenic disorder which taken with the authors results and inclusion of all SNPs in the genome suggests different PRS liability and thresholds could impact their results. How are the PRS-protein results impacted when different discovery thresholds are used (e.g. $P < 0.5$, $P < 0.25$, $P < 0.1$, $P < 0.01$)?"

We thank the reviewer for raising this interesting point. As a result, we have completed a correlational analysis examining all discovery threshold ($P < 0.01$, $P < 0.05$, $P < 0.1$, $P < 0.5$ and $P < 1$) versus Olink® neurology protein levels. We find that results are consistent across each threshold with no significant correlations following multiple testing correction. Results are shown for each phenotype in the Supplementary Polygenic Risk Score Analysis file. The following text has been added to methods section 4.5 on lines 435-437:

"Results were consistent across all discovery thresholds ($P < 0.01$, $P < 0.05$, $P < 0.1$, $P < 0.5$ and $P < 1$; see Supplementary Polygenic Risk Score Analysis)."

Comment 2.3

"The authors state that they adjusted for cell composition in their EWAS analysis using basophils, eosinophils, neutrophils, lymphocytes, and monocytes cell count information. I am concerned that one of the main results, "The majority of the protein-CpG associations were attributable to CRTAM ... which is upregulated in CD4+ and CD8+ cells" is driven by differences in these cell types that were not captured in the cell type adjustment. The authors should use the Houseman method (PMID: 22568884) to specifically compute the proportion of CD4+ and CD8+ cell types and include them as covariates in their analytic models."

The authors thank the reviewer for this suggestion. As a result, we have used the Houseman method to compute the proportion of white blood cell types in blood samples and have used these as covariates in our analytic models. Following this adjustment, we have substantially updated the results in sections 2.3 accordingly, as well as Figures 3 and 4.

"For the EWAS, a Bonferroni P value threshold of 3.9×10^{-10} (genome-wide significance level: $3.6 \times 10^{-8}/92$ proteins) was set (Saffari et al., 2018). We identified 26 genome-wide significant CpG sites associated with the levels of 9 neurological proteins ($P < 3.9 \times 10^{-10}$). Of these associations, 17 were cis effects (35.6%) and 9 associations were trans effects (65.4%; with 6 trans variants located on chromosomes distinct for their respective Olink gene) (Figure 3; Supplementary Table 6)."

Comment 2.4

“When testing for colocalization of SNPs driving multiple associations, the authors state they selected a “200kb region surrounding the sentinel cis pQTL”. Because physical distance does not capture LD structure, the authors should instead consider selection of SNPs to test for colocalization based on LD block as opposed to absolute physical distance.”

We appreciate this concern and see the validity of the point raised. We elected this approach as per the method recommended by the developers of the ‘coloc’ package who originally employed this 200 kb region (Guo et al., 2015).

Comment 2.5

“One of the major innovative features of the study that is highlighted by the authors is the integration of data across omics types. However, much of this type of analysis is carried out using expression databases as opposed to testing DNA Methylation Outcome relationships using data they collected from the same study participants. Am I missing the rationale for why DNA methylation could be mediating the genetic-protein level/disease associations? Could you perform statistical analyses to see if methylation is mediating the genetic protein level associations?”

To probe molecular mechanisms underlying altered Olink® proteins, we incorporated methylation data to further determine whether genetic variation affects transcriptional events. However, we acknowledge that we failed to formally test this, by only showing whether pQTLs were previously identified as methylation QTLs (mQTLs), which may suggest that altered DNA methylation affects protein levels. To address this shortcoming, we have performed Mendelian randomisation using mQTLs (from Phenoscanner) and pQTLs (from our study) as instrumental variables. Interestingly, we have found that while differential DNA methylation is causally associated with altered protein levels (of MATN3, MDGA1 and NEP), altered protein levels are reciprocally associated with changes in DNA methylation. These novel results are presented in full in Supplementary Table 8. A full Mendelian randomisation methods section has been added (Methods section 4.14). Furthermore, the following text has been added to Results section 2.3 on lines 203-208:

“We performed bidirectional MR analyses to formally test whether there was a causal relationship between DNA methylation at these sites and Olink® protein levels (see methods). For each protein, MR analyses suggested that differential DNA methylation was causally associated with changes in protein levels. Conversely, altered protein levels of MATN3, MDGA1 and NEP were also causally associated with differential methylation levels at CpG sites identified by our EWAS (Supplementary Table 8).”

Comment 2.6

"It is unclear on line 148, which pruned cis QTLs were instrumental variables? Was it pQTLs, eQTLs?"

We thank the reviewer for highlighting the ambiguous use of language in this portion of the manuscript. As this was a bidirectional Mendelian randomisation analysis, *cis* eQTLs were firstly used as instrumental variables to investigate whether gene expression was causally linked to changes in protein levels. Conversely, *cis* pQTLs were used as instrumental variables to investigate whether protein levels were causally associated with changes in gene expression. The following text has been added to lines 163-164:

"Pruned cis protein and expression QTL variants (linkage disequilibrium (LD) $r^2 < 0.1$) were used as instrumental variables for the bidirectional MR analyses."

Comment 2.7

"It is unclear how many samples and SNPs were filtered out at each quality control step or how many samples and loci were tested in each analysis. It would also be beneficial to understand the number of individuals that contributed to both the GWAS and EWAS analyses. Please provide a flow chart and/or report quality control results in the results/methods section."

We thank the reviewer for this suggestion. A flow chart of quality control and exclusion criteria has been created and included in Supplementary Data 3, which has been referenced to on the text on lines 385-386:

"Key inclusion/exclusion criteria for the present study are highlighted in Supplementary Data 3."

Comment 2.8

"The authors state that sex, age, and genetic ancestry were adjusted for at the protein measurement level and the residuals were used in downstream EWAS and GWAS (described in section 4.2). What is the rationale for adjusting for genetic ancestry at the protein level as opposed to in the GWAS statistical models? In section 4.7, the epigenome analyses, sex and age appear to be included in the EWAS models as a covariate, it isn't clear to me why these would be adjusted for again?"

Firstly, genetic ancestry was adjusted for at the protein measurement level for computational efficiency when running the GWAS. Secondly, age and sex were adjusted for in the EWAS models as methylation (CpG site) was the dependent variable in these models and merited such adjustment. The text in section 4.7 has been updated accordingly:

“Epigenome-wide association analyses were conducted by regressing each of 459,309 CpG sites (as dependent variables) on transformed protein levels using linear regression with adjustments for age, sex, estimated white blood cell proportions (CD4⁺ T cells, CD8⁺ T cells, B cells, Natural Killer cells, granulocytes and monocytes) and technical covariates (plate, position, array, hybridisation, date). White blood cell proportions were estimated from methylation data using the Houseman method (Houseman et al., 2012).”

Comment 2.9

“The methods section describing GWAS imputation seems to fit better further up near the description of the genotyping measurements and before the polygenic risk score description.”

This section has been moved to the appropriate location as per the recommendation of the reviewer.

Comment 2.10

“There is no description of how genetic ancestry was ascertained.”

We thank the reviewer for highlighting this omission. To correct this, the following text has been added to lines 401-403:

“To obtain an estimate of population structure, multidimensional scaling (MDS) was performed on LBC1936 genotyping data and the first four MDS components were used to control for genetic ancestry in the analytic models.”

Comment 2.11

“The description of GO enrichment tests does not specify what the null set of sites included. For example, was it all loci in the genome, all loci measured on the 450K, etc? It should be based on the set of measured/assayed genomic loci/genes. Also, a description of how CpG sites were mapped to genes for GO testing is needed.”

To address this omission, we would firstly like to clarify that the null set of sites included all annotated gene symbols from the 450K array. This omission has been rectified in the text by the following addition to methods section 4.7 on lines 459-463:

“All gene symbols from the 450K array annotation (null set of sites) were converted to Entrez IDs using biomaRt (Durinck et al., 2005, Durinck et al., 2009). GO terms and their corresponding gene sets were obtained from the Molecular Signatures Database (MSigDB)-Collection 5 (Liberzon et al., 2015) while KEGG pathways were downloaded from the KEGG REST server (Tenenbaum, 2016).”

Comment 2.12

“Statistical models used for GWAS and EWAS should be specified”

In order to address this comment, the models for GWAS and EWAS have been added to sections 4.6 and 4.7 respectively:

GWAS model: Olink® protein residuals ~ SNP

EWAS model: CpG ~ Olink® protein residuals + age + sex + estimated white blood cell proportions + plate + array + date + set + position

We would also like to highlight that an additional EWAS analyses of protein levels was performed using **OmicS**-data-based **Complex trait Analysis** software (OSCA; mixed-linear-model based method) along with limma (linear-model based method). We carried this out as an additional analysis as OSCA has been shown to identify less spurious associations when compared to other EWAS methodologies (including linear regression). We found a strong overlap in CpG sites identified by both methods (82.6%). Output from the OSCA models has been included in Supplementary Table 7. We have added in the following text under results section 2.3 and methods section 4.8:

Updated Results - Lines 180-189:

*“As an additional analysis, we performed a mixed-linear-model approach to perform EWAS termed OSCA (**OmicS**-data-based **Complex trait Analysis**)-MOMENT. OSCA has been recently shown to identify less spurious signals than other methods (including linear regression) (Zhang et al., 2019). Of the 9 proteins with genome-wide significant CpG sites identified using limma (n = 26 CpG sites), 8 proteins were also shown to have genome-wide significant associations using OSCA (n = 23 CpG sites; 14 cis (60.9%) and 9 trans (39.1%) associations). Indeed, only CRTAM failed to show a Bonferroni-corrected significant association using OSCA when compared to limma. Furthermore, of the 23 CpG sites identified using OSCA, 19/23 CpGs (82.6%) were also identified by EWAS performed using limma showing a strong overlap between both methods (Supplementary Table 7).”*

Updated Methods - Lines 469-478:

“We also performed EWAS analyses of Olink protein levels using OmicS-data-based Complex trait Analysis software (OSCA). We carried out OSCA as an additional EWAS analysis as it has recently been shown to identify less spurious associations when compared to other methods (including linear regression) (Zhang et al., 2019). CpG site was the dependent variable whereas Olink protein levels were input as independent variables. Models were adjusted for age, sex, estimated white blood cell proportions (CD4⁺ T cells, CD8⁺ T cells, B cells, Natural Killer Cells, granulocytes and monocytes) and technical covariates (plate, position, array, hybridisation, date) as in section 4.7. The MOMENT method was used to test for associations between traits of interest and DNAm at individual probes. MOMENT is a mixed-linear-model based method that can account for unobserved confounders and the correlation between distal probes which may be introduced by such confounders.”

Comment 2.13

“There is no description of the Mendelian Randomization analyses in the method section”

We agree that a methods section for Mendelian Randomisation (MR) should be included given that MR is now used for three separate analyses in the manuscript: to test for causal relationships between PVR and AD, gene expression and protein levels as well as methylation and protein levels. The following section has been added as methods section 4.14:

“Two-sample bidirectional Mendelian Randomisation was used to test for putatively causal relationships between (i) PVR, a cell-surface glycoprotein, and AD risk, (ii) gene expression and plasma protein levels and (iii) DNA methylation and plasma protein levels. Pruned variants ($LD r^2 < 0.1$) were used as instrumental variables (IV) in MR analyses. In cases where only one independent SNP remained after LD pruning, causal effect estimates were determined using the Wald ratio test. When multiple independent variants were present, and if no evidence of directional pleiotropy was present (non-significant MR-Egger intercept), multi-SNP MR was conducted using inverse variance-weighted estimates. All MR analyses were conducted using MRbase (Hemani et al., 2018).

(i) Either pQTLs for PVR levels identified in this study or genome-wide significant SNPs from a GWAS on AD risk were used as the appropriate IV in the bidirectional MR analysis (Jansen et al., 2019). While 72 genome-wide significant cis pQTLs were identified for PVR levels, only one SNP (rs7255066) remained after LD pruning. Five independent SNPs were

identified and used as IV to test for a causal relationship between AD risk and altered plasma PVR levels.

(ii) Expression QTLs obtained from eQTLGen consortium were used as IV to test whether changes in gene expression were causally associated with protein levels (Võsa et al., 2018). Protein QTLs identified by our GWAS were as IV to test whether protein levels were causally associated with altered gene expression.

(iii) For the three proteins with GWAS and EWAS associations (MATN3, MDGA1 and NEP), we wished to test whether methylation affected protein levels and/or whether protein levels affected methylation. We queried Phenoscanner to examine whether pQTLs for protein levels of MATN3, MDGA1 and NEP, identified in this study, were previously identified as methylation QTLs (mQTLs) for corresponding genome-wide significant CpG sites (Staley et al., 2016). Methylation QTLs were used as IV to test whether changes in DNA methylation were causally associated with Olink® protein levels. Conversely, pQTLs were used as IV to determine whether altered protein levels were causally linked to differential methylation levels. Of note, as methylation of the 11 cis CpG sites associated with differential MDGA1 levels in our study are highly inter-correlated (Supplementary Data 6), we considered only the most significant cis CpG site (cg20053110) for MR analyses.”

Comment 2.14

“The authors seem to have used some outdated summary statistics for PRS or did not specify the exact source of the summary statistics (e.g. autism)”

We thank the reviewer for highlighting this concern. Every effort was made by the authors to obtain the most up-to-date summary statistics for PRS; however, in certain cases, the summary statistics were not available following attempts to contact relevant GWAS study authors. The source of summary statistics for autism, and all phenotypes, have been further clarified in Supplementary Table 16.

References

- GUO, H., FORTUNE, M. D., BURREN, O. S., SCHOFIELD, E., TODD, J. A. & WALLACE, C. 2015. Integration of disease association and eQTL data using a Bayesian colocalisation approach highlights six candidate causal genes in immune-mediated diseases. *Human molecular genetics*, 24, 3305-3313.
- HEMANI, G., ZHENG, J., ELSWORTH, B., WADE, K. H., HABERLAND, V., BAIRD, D., LAURIN, C., BURGESS, S., BOWDEN, J., LANGDON, R., TAN, V. Y., YARMOLINSKY, J., SHIHAB, H. A., TIMPSON, N. J., EVANS, D. M., RELTON, C., MARTIN, R. M., DAVEY SMITH, G., GAUNT, T. R. & HAYCOCK, P. C. 2018. The MR-Base platform supports systematic causal inference across the human phenome. *Elife*, 7.
- HOUSEMAN, E. A., ACCOMANDO, W. P., KOESTLER, D. C., CHRISTENSEN, B. C., MARSIT, C. J., NELSON, H. H., WIENCKE, J. K. & KELSEY, K. T. 2012. DNA methylation arrays as surrogate measures of cell mixture distribution. *BMC Bioinformatics*, 13, 86.

- JANSEN, I. E., SAVAGE, J. E., WATANABE, K., BRYOIS, J., WILLIAMS, D. M., STEINBERG, S., SEALOCK, J., KARLSSON, I. K., HAGG, S., ATHANASIU, L., VOYLE, N., PROITSI, P., WITOELAR, A., STRINGER, S., AARSLAND, D., ALMDAHL, I. S., ANDERSEN, F., BERGH, S., BETTELLA, F., BJORNSSON, S., BRAEKHUS, A., BRATHEN, G., DE LEEUW, C., DESIKAN, R. S., DJUROVIC, S., DUMITRESCU, L., FLADBY, T., HOHMAN, T. J., JONSSON, P. V., KIDDLE, S. J., RONGVE, A., SALTVEDT, I., SANDO, S. B., SELBAEK, G., SHOAI, M., SKENE, N. G., SNAEDAL, J., STORDAL, E., ULSTEIN, I. D., WANG, Y., WHITE, L. R., HARDY, J., HJERLING-LEFFLER, J., SULLIVAN, P. F., VAN DER FLIER, W. M., DOBSON, R., DAVIS, L. K., STEFANSSON, H., STEFANSSON, K., PEDERSEN, N. L., RIPKE, S., ANDREASSEN, O. A. & POSTHUMA, D. 2019. Genome-wide meta-analysis identifies new loci and functional pathways influencing Alzheimer's disease risk. *Nat Genet*.
- VÖSA, U., CLARINGBOULD, A., WESTRA, H.-J., BONDER, M. J., DEELEN, P., ZENG, B., KIRSTEN, H., SAHA, A., KREUZHUBER, R., KASELA, S., PERVJAKOVA, N., ALVAES, I., FAVE, M.-J., AGBESSI, M., CHRISTIANSEN, M., JANSEN, R., SEPPÄLÄ, I., TONG, L., TEUMER, A., SCHRAMM, K., HEMANI, G., VERLOUW, J., YAGHOOTKAR, H., SÖNMEZ, R., ANDREW, A. A., KUKUSHKINA, V., KALNAPENKIS, A., RÜEGER, S., PORCU, E., KRONBERG-GUZMAN, J., KETTUNEN, J., POWELL, J., LEE, B., ZHANG, F., ARINDRARTO, W., BEUTNER, F., BRUGGE, H., DMITRIEVA, J., ELANSARY, M., FAIRFAX, B. P., GEORGES, M., HEIJMANS, B. T., KÄHÖNEN, M., KIM, Y., KNIGHT, J. C., KOVACS, P., KROHN, K., LI, S., LOEFFLER, M., MARIGORTA, U. M., MEI, H., MOMOZAWA, Y., MÜLLER-NURASYID, M., NAUCK, M., NIVARD, M., PENNINX, B., PRITCHARD, J., RAITAKARI, O., ROTZSCHKE, O., SLAGBOOM, E. P., STEHOUWER, C. D. A., STUMVOLL, M., SULLIVAN, P., 'T HOEN, P. A. C., THIERY, J., TÖNJES, A., VAN DONGEN, J., VAN ITERSOM, M., VELDINK, J., VÖLKER, U., WIJMENGA, C., SWERTZ, M., ANDIAPPAN, A., MONTGOMERY, G. W., RIPATTI, S., PEROLA, M., KUTALIK, Z., DERMITZAKIS, E., BERGMANN, S., FRAYLING, T., VAN MEURS, J., PROKISCH, H., AHSAN, H., PIERCE, B., LEHTIMÄKI, T., BOOMSMA, D., PSATY, B. M., GHARIB, S. A., AWADALLA, P., MILANI, L., OUWEHAND, W. H., DOWNES, K., STEGLE, O., BATTLE, A., YANG, J., VISSCHER, P. M., SCHOLZ, M., GIBSON, G., ESKO, T. & FRANKE, L. 2018. Unraveling the polygenic architecture of complex traits using blood eQTL meta-analysis. *bioRxiv*, 447367.
- ZHANG, F., CHEN, W., ZHU, Z., ZHANG, Q., DEARY, I. J., WRAY, N. R., VISSCHER, P. M., MCRAE, A. F. & YANG, J. 2019. OSCA: a tool for omic-data-based complex trait analysis. *bioRxiv*, 445163.

Reviewers' Comments:

Reviewer #1:

Remarks to the Author:

The authors adequately responded to the comments of both reviewers. The only remaining comment I would have is that the abstract might benefit from adding a bit more detail. For instance, the authors write that "Using this information, we identified biological pathways in which putative neurological biomarkers are implicated as well as molecular mechanisms through which genetic variation may perturb plasma protein levels." and then continue with a conclusion that is mostly a repetition of information already provided a few lines above. Instead the authors may want to be a bit more specific about the pathways they identified to raise the reader's interest.

Reviewer #2:

Remarks to the Author:

I'd like to thank the authors for sufficiently addressing nearly all of my previous concerns; the manuscript is much improved. However, there is still one concern, regarding the polygenic risk score (PRS) analysis, that I do not think was sufficiently addressed and raises serious doubts about one of the authors findings. I have detailed the points of concern below, in case the authors decide to keep this section of the manuscript, but actually think the paper is strong and impactful without the PRS analyses. Therefore, I suggest the authors consider simply removing the PRS-based work from the manuscript to make it suitable for publication.

1. As now detailed in new Supplementary Table 16, the authors used seriously outdated GWAS results for at least a few of the initial phenotypes that I spot checked (ASD and ADHD) and in their response to reviewers they stated that:

"Every effort was made by the authors to obtain the most up-to-date summary statistics for PRS; however, in certain cases, the summary statistics were not available following attempts to contact relevant GWAS study authors."

This response didn't make sense to me after seeing that the PGC download webpage was listed as the source of their GWAS summary statistics for ASD and ADHD. Since the 2017-2019 GWAS summary statistics (at least for those 2 outcomes I initially spot checked) have been and are still readily available (I double checked and downloaded myself recently), I am puzzled why the latest statistics could not be obtained by the authors – they do not need to be obtained from the author of a manuscript.

When computing PRS for these outcomes it is critical that the most recent version is used because the sample sizes are wildly different and the summary statistic weights and p-values being used are inaccurate due to the much smaller sample size of the study the authors pulled these statistics from. Specifically, the GWAS summary statistics the authors use in this manuscript to generate ASD and ADHD PRS examined a small number of trios (~2,000 or less) and less than 1,000 cases and controls combined. In comparison, the 2017 GWAS summary statistics available for download from the PGC website examined over 98,000 ADHD cases and 1.9 million controls. Similarly, for ASD, the 2017 and 2019 summary statistics were derived from ~50,000 ASD cases and controls. This will result in incorrect weighting of the SNPs used to derive the PRS and, thus, an incorrect PRS. I have provided these 2 outcomes as an example but the authors should check all of the phenotypes and make sure the most up to date summary statistics are used

2. My second concern with associating PRS with the circulating biomarkers is that the population under examination likely is not representative of the full spectrum of scores for many of these outcomes, i.e. the range of the scores is probably mostly distributed in the "normal" range

because there are a small number of affected individuals for several of these outcomes (e.g. schizophrenia, ASD, etc). If the point of the PRS correlation analysis is to determine whether any of the brain biomarkers are related to differences in genetic risk liability for particular neurological brain disorders, you would not expect to see correlations if there are too few individuals in your population with the outcome of interest; the PRS score distributions for many of these outcomes are likely not capturing affected individuals so are not representative/capturing genetic risks for those outcomes per se.

Response to Reviewers

We are very grateful for the comments provided by the editor and each of the external reviewers of this manuscript. We are pleased that the reviewers feel their comments have been adequately addressed. Please see below, in blue, a detailed response to the remaining comments and concerns raised by the reviewers. Line numbers refer to the manuscript file with highlighted changes.

Response to the reviewers

Reviewer #1

"The only remaining comment I would have is that the abstract might benefit from adding a bit more detail. For instance, the authors write that "Using this information, we identified biological pathways in which putative neurological biomarkers are implicated as well as molecular mechanisms through which genetic variation may perturb plasma protein levels." and then continue with a conclusion that is mostly a repetition of information already provided a few lines above. Instead the authors may want to be a bit more specific about the pathways they identified to raise the reader's interest".

We thank the reviewer for this comment and agree that the abstract would benefit from more specific detail and less repetition. To rectify this, we have modified the abstract from lines 41-46 as follows:

Using this information, we identify biological pathways in which putative neurological biomarkers are implicated (neurological, immunological and extracellular matrix metabolic pathways). We also observe causal relationships between changes in gene expression (DRAXIN, MDGA1 and KYNU), or DNA methylation profiles (MATN3, MDGA1 and NEP), and altered plasma protein levels. Together, this may help inform causal relationships between biomarkers and neurological diseases."

Reviewer #2

"However, there is still one concern, regarding the polygenic risk score (PRS) analysis, that I do not think was sufficiently addressed and raises serious doubts about one of the authors findings. I have detailed the points of concern below, in case the authors decide to keep this section of the manuscript, but actually think the paper is strong and impactful without the PRS analyses. Therefore, I suggest the authors consider simply removing the PRS-based work from the manuscript to make it suitable for publication."

We thank the reviewer for their comments regarding the polygenic risk score (PRS) analyses. Upon reflection of these comments and concerns, we agree that the PRS analyses should be removed from the manuscript and do not add to the overall strength of the paper. As a result, we have made the following modifications to omit the PRS analyses from the study:

Results section 2.4 on Lines 221-230, detailing the PRS analyses, has been deleted. Furthermore, Supplementary Tables 12 and 16 have been removed with the remaining Supplementary Tables/Data files renumbered to reflect this change. We have also removed Methods Section 4.5, on lines 421-426, which details the calculation of PRS. Finally, the supplementary file "Supplementary Polygenic Risk Score Analysis" may also be removed from the system.